# Connectome-constrained latent variable models of whole-brain neural activity

**Lu Mi**[1,3], **Richard Xu**[1], **Sridhama Prakhya**[1], **Albert Lin**[2], **Nir Shavit**[3],
**Aravinthan D.T. Samuel**[2†], **Srinivas C. Turaga**[1†]
[1] HHMI Janelia Research Campus
[2] Harvard University
[3] MIT
{xur,prakhyas,turagas}@janelia.hhmi.org
{albertlin,samuel}@g.harvard.edu
{lumi,shanir}@mit.edu

## Abstract

The availability of both anatomical connectivity and brain-wide neural activity measurements in *C. elegans* make the worm a promising system for learning detailed, mechanistic models of an entire nervous system in a data-driven way. However, one faces several challenges when constructing such a model. We often do not have direct experimental access to important modeling details such as single-neuron dynamics and the signs and strengths of the synaptic connectivity. Further, neural activity can only be measured in a subset of neurons, often indirectly via calcium imaging, and significant trial-to-trial variability has been observed. To address these challenges, we introduce a connectome-constrained latent variable model (CC-LVM) of the unobserved voltage dynamics of the entire *C. elegans* nervous system and the observed calcium signals. We used the framework of variational autoencoders to fit parameters of the mechanistic simulation constituting the generative model of the LVM to calcium imaging observations. A variational approximate posterior distribution over latent voltage traces for all neurons is efficiently inferred using an inference network, and constrained by a prior distribution given by the biophysical simulation of neural dynamics. We applied this model to an experimental whole-brain dataset, and found that connectomic constraints enable our LVM to predict the activity of neurons whose activity were withheld significantly better than models unconstrained by a connectome. We explored models with different degrees of biophysical detail, and found that models with realistic conductance-based synapses provide markedly better predictions than current-based synapses for this system.

## 1 Introduction

The anatomical connectivity of the entire *C. elegans* nervous system, including both chemical and electrical synapses, has been known for several decades [30; 27; 31]. However, well-calibrated and predictive connectome-constrained mechanistic models of this nervous system have yet to be demonstrated [13; 28; 10; 7]. This is partially because whole-brain recordings are insufficient to completely constrain computational models. First, the single-neuron and synapse dynamics are generally unknown. Second, the connectome does not directly inform the signs and strengths of individual synapses. Third, the response properties of sensory neurons are incompletely known. Further, it is unclear what level of biophysical detail is necessary to reproduce the essential computational function of the *C. elegans* nervous system.

Here, we used whole-brain calcium imaging data to constrain the missing parameters in a connectome-constrained, biophysically detailed model of the *C. elegans* nervous system. We started with a simplified non-spiking passive point-neuron model of the voltage dynamics of individual neurons in the circuit. We modeled inputs to the neurons from both electrical and chemical synapses.

---

[†]equal contribution

The chemical synapses are modeled nonlinearly, with either current-based or conductance-based biophysics. The challenge of fitting such a model to data is two-fold. First, the voltage dynamics of the neurons are not directly observed, but rather indirectly measured via their slow calcium dynamics. Second, there is significant trial-to-trial variability in neural activity, suggesting strong initial state dependence in the neural responses to the same sensory stimulus [9]. These issues can both be addressed by treating the collective voltage signals of all neurons in the nervous system as an unobserved latent variable whose dynamics are determined by the simplified connectome-constrained biophysical model with unknown neuronal and synaptic parameters.

Our connectome-constrained latent variable model (CC-LVM) of voltage dynamics of the *C. elegans* nervous system is a large-scale latent variable model with a very high-dimensional latent space consisting of voltage dynamics of 300 neurons over 5 minutes of time at the simulation frequency of 160 Hz. The generative model for these latent variables is described by stochastic differential equations modeling the nonlinear dynamics of the network activity. We developed a variational autoencoder-based framework for inferring the unobserved voltage dynamics from the observed calcium dynamics of only a subset of the neurons in the nervous system.

The *C. elegans* nervous system consists of 300 neurons, divided into 118 distinct classes (often bilaterally symmetric pairs) [26; 14; 25]. These neurons can be categorized into sensory neurons, interneurons, and motor neurons. The majority of these neurons, about 200, are concentrated in the head of the animal. The synaptic connectivity of *C. elegans* has been mapped with electron microscopy, providing researchers with a complete connectome [30; 31].

We applied the CC-LVM to whole-brain calcium imaging data which captured 170 of the 300 neurons in multiple worms as they responded to chemosensory stimuli [32]. In principle, an accurate model of the nervous system constrained by incomplete activity measurements but a complete description of the animal's anatomical connectivity can enable accurate predictions of activity in neurons which were not recorded. We tested this hypothesis by using the CC-LVM to predict the activity of neurons which were measured but whose activity was withheld during model training. We also used the model to predict the activity of entire worms, by holding out single trials during training. The CC-LVM predicted activity of withheld neurons and withheld worms significantly better than models unconstrained by the connectome, demonstrating the utility of the connectome even when little is known about the signs and strengths of individual connections. Further, we found that models with conductance-based synapses provide superior predictions to models with current-based synapses. This is surprising because while conductance-based synapses are more mechanistically accurate, they do not necessarily support efficient inference [8]. CC-LVMs thus provide a new tool for connectome and activity constrained modeling of neural circuits and for discovering the appropriate level of detail of biophysical model for a given system. This work provides a framework not only for modeling the *C. elegans* nervous system in particular, but also the neuronal networks of other biological systems.

## 1.1 Prior Work

Previous work in modeling the *C. elegans* nervous system has focused on creating network simulations based on anatomical connectome data with unknown single neuron synaptic biophysics [18; 7; 28; 19; 13]. These network models provide a holistic view of the entire nervous system and were validated by comparing simulated locomotion with movements of live animals [11]. However, they were not able fit against prerecorded calcium fluorescence data, so their simulated neuronal activities were not confirmed. Work has also been done on modeling unlabeled neuron populations but these models cannot predict activity at a single neuron resolution because the activity data is not mapped to specific cells [4]. Recent advances in machine learning have enabled increasingly sophisticated latent variable models that uncover structure from data generated by advanced neural interfacing technologies [21]. This framework has been used in other contexts to infer neuronal voltage dynamics from high-dimensional calcium fluorescence recordings [20; 1; 24]. Despite their utility, these models do not account for connectomic data. Our work employs both LVMs and connectomic constraints to create a model that is informed both by neural population dynamics and anatomical network structure.

Recently, Bayesian models have been widely applied in neuronal datasets to infer variables such as spiking activity, neural dynamics, and connectivity [28; 1; 24]. In particular, Linderman et al. [17] created a hierarchical state space model of *C. elegans* neural activity. However, because such a model is not mechanistic, it cannot predict the activity of neurons that were never measured. Warrington

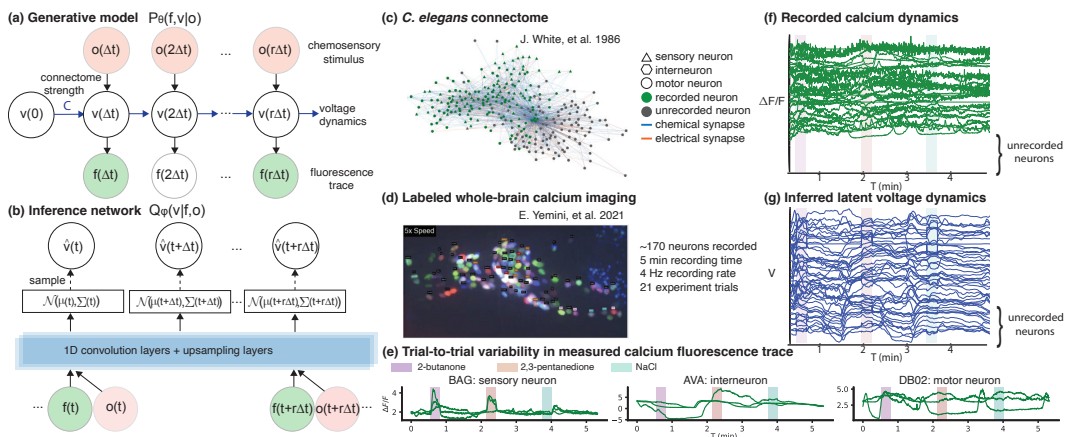

Figure 1: **Anatomical and functional whole-brain datasets available in *C. elegans*. a.** The connectome-constrained generative model produces a prior distribution of voltage dynamics and fluorescence traces given chemosensory stimulus. Observed data is highlighted in color and $r$ is the upsampling factor from the acquisition rate to the simulation rate. **b.** The inference network generates a posterior distribution of voltage dynamics given observed fluorescence traces and chemosensory stimuli. **c.** The connectome of *C.elegans* nervous system, with chemical connections in blue and electrical connections in orange [30]. Neurons recorded in the functional imaging dataset [32] are colored green. **d.** Labeled whole-brain calcium imaging of 170 neurons in the head of the worm while a panel of chemosensory stimuli were presented [32]. Deterministic multicolor labels allow for the identification of all neurons. **e.** Sample fluorescence activity traces of three measured neurons, with stimulus delivery marked by the colored bars. Note the significant trial-to-trial variability in neural activity.

et al. [28] used sequential Monte Carlo (SMC) to impute the intracellular voltage potentials of *C. elegans* neurons from 49 recorded calcium traces [12]. Their work combined neuronal, body and calcium observation simulators in order to model locomotion [3]. Their simulator produced a series of exemplar neuronal voltage traces, but they did not test their model's ability to predict the activity of unrecorded neurons, so the accuracy of their inferences remains unknown.

Our model uses a variational auto-encoder (VAE) to perform inference instead of SMC which is more computationally efficient because it allows direct sampling from the approximate posterior. The CC-LVM was trained on activity recordings of 170 neurons [32], a large fraction of the 300 total neurons in the *C. elegans* nervous system [29; 31]. We validated our model by evaluating its ability to predict the activity of neurons held out from the training data. Finally, we searched a space of generative models and optimized voltage predictions by comparing them to calcium activity data.

## 2   CONNECTOME-CONSTRAINED LATENT VARIABLE MODEL

We constructed a connectome-constrained latent variable model (CC-LVM) of the *C. elegans* nervous system, where each node in our network represents a specific neuron in the animal. The activity of each neuron was modeled with a latent variable analogous to voltage. Since the neurons in *C. elegans* do not spike [2], the dynamics of each neuron in the network is represented by a stochastic non-spiking passive leaky integrator equation with learned time constants and resting membrane potentials [5]. The neurons were coupled by both chemical and electrical synapses with learned weights [5]. We allowed these weights to be non-zero only where the connectome indicates the existence of synapses. Given a set of learned parameters (weights, time constants, and resting membrane potentials), the dynamics of the network define a prior distribution over neural activity trajectories. Importantly, the stochastic nature of the neuron voltages causes deviations from perfectly deterministic dynamics, allowing us to model the observed variability in single-neuron dynamics. This variability has several potential sources, including the unmeasured initial states of the neurons and our incomplete knowledge of the sensory inputs driving the nervous system.

A latent variable model of this scale with a nonlinear generative model defined by the stochastic dynamics is difficult to fit because of the high number of parameters. To address this challenge, we used the probabilistic inference framework of variational autoencoders (VAE) [22] to train a black-box voltage inference network to predict a posterior distribution over neural activity trajectories. We then used this inference network to train the parameters of the CC-LVM. The resulting LVM has a biologically realistic generative model of the nonlinear neural dynamics of the *C. elegans* nervous system, and a black-box temporal convolutional inference network which, given sensory stimulus

and calcium imaging measurements, predicts a factorized Gaussian distribution over the voltages of all the neurons in the network. Several variants of the LVM were developed: we tested models in which the synaptic connections were modeled as either *current-based* or *conductance-based* [5], and evaluated different types of connectome constraint. The LVM was optimized with the ELBO (evidence lower bound) objective.

## 2.1 NETWORK MODEL WITH PASSIVE POINT-NEURON VOLTAGE DYNAMICS

Neurons in the *C. elegans* nervous system are largely non-spiking [2], so we model the voltage dynamics for these neurons as a passive point neurons with a single electrical compartment. Let $\boldsymbol{v} \in \mathbb{R}^N$ denote the voltages of the $N$ neurons. The voltage $v_i(t)$ for each post-synaptic neuron $i$ at the time $t$ was calculated using a first-order leaky integrator equation given by

$$\tau_i \dot{v}_i(t) + v_i(t) = s_i^c(t) + s_i^e(t) + v_i^{rest} + o_i(t), \tag{1}$$

where $\tau_i$ is the voltage time constant, $o_i$ is the chemosensory input provided to only the sensory neurons, $s_i^c$ is the chemical synaptic input, $s_i^e$ is the electrical synaptic input, $v_i^{rest}$ is the resting neuron voltage.

We studied two variations of the model, the *current-based model* and the *conductance-based model*, which differ in their formulations of chemical synaptic input $s_i^c$. Since neurons in the *C. elegans* nervous system are largely non-spiking, we model the chemical synapses as having graded release of neurotransmitter, rather than the all-or-none quantal release seen in spiking neurons. In both models, we model the amount of neurotransmitter released, $W_{ji}^c g(v_j(t))$, in proportion to the pre-synaptic voltage $v_j$ followed by a softplus activation $g(\cdot)$ which sets a minimum voltage below which there is no synaptic release. We use $g(\cdot)$, to denote a softplus function for the rest of the paper. In our current-based model, synaptic input $s_i^c$ to a post-synaptic neuron $i$ is directly proportional pre-synaptic neurotransmitter concentration:

$$s_i^c(t) = \sum_j^N W_{ji}^c g(v_j(t)), \tag{2}$$

where $W_{ji}^c$ represents the chemical synaptic weight between pre-synaptic neuron $j$ and post-synaptic neuron $i$. $W_{ji}^c$ can be positive or negative depending on if the synaptic connection is excitatory or inhibitory. If neurons $j$ and $i$ are not connected, $W_{ji}^c$ is set to zero. In the conductance-based model, we model the synaptic current entering the post-synaptic neuron with more biophysical detail as

$$s_i^c(t) = \sum_j^N (E_{ji} - v_i(t)) W_{ji}^c g(v_j(t)). \tag{3}$$

Here, the pre-synaptic neurotransmitter concentration $W_{ji}^c g(v_j(t))$ is more accurately modeled as proportional to the conductance at the post-synaptic terminal. The post-synaptic current is then given by the product of the synaptic conductance and the difference between the post-synaptic voltage $v_i(t)$ and the synaptic reversal potential $E_{ji}$.

In contrast to the current-based synapse, whose input is independent of the post-synaptic voltage $v_i(t)$, the more biophysically accurate conductance-based synapse model also has a dependence on the post-synaptic voltage. Additionally, this model decouples the sign of the synapse from the strength. The reversal potential of a synapse $E_{ji}$ dictates whether a synapse is excitatory or inhibitory. A large and positive $E_{ji}$ corresponds to an excitatory synapse causing depolarization of the postsynaptic neuron. Conversely, an inhibitory synapse will have a negative $E_{ji}$ causing hyperpolarization. In this model, we can now independently train the sign of a synapse and its non-negative strength $W_{ji}^c$, which is not easily possible with current-based synapses. In both the current-based and conductance-based models, the following equation was used to represent electrical synaptic inputs:

$$s_i^e(t) = \sum_j^N W_{ji}^e (v_j(t) - v_i(t)), \tag{4}$$

where $W_{ji}^e$ is restricted to be non-negative and $v_j - v_i$ is the potential difference between presynaptic and postsynaptic neurons. We also restrict $W_{ji}^e = W_{ij}^e$ because the potential differences between electrical synapses are symmetric. To directly compare the outputs of the LVM to neural activity

measurements, our model must generate calcium signals from the voltage traces. We model the calcium concentration $[\text{Ca}]_i$ of each neuron $i$ as a first-order leaky integrator, driven by voltage-gated calcium channels with the same nonlinear current-voltage (I-V) function $g(v_i)$:

$$\tau_{[\text{Ca}]}[\dot{\text{Ca}}]_i(t) + [\text{Ca}]_i(t) = g(v_i(t)), \tag{5}$$

where $g(\cdot)$ represents softplus activation, $\tau_{[\text{Ca}]}$ is a time constant shared across all neurons. We then map calcium concentration $[\text{Ca}]$ into the measured calcium fluorescence signals $\boldsymbol{f}$ via an affine transform with scalar $\alpha^f$ and bias $\beta^f$,

$$f_i(t) = \alpha^f[\text{Ca}]_i(t) + \beta^f + \sigma^f\epsilon_i^f(t), \tag{6}$$

with measurement noise represented by a noise amplitude $\sigma_f$ and a noise term $\epsilon_i^f(t) \sim \mathcal{N}(0,1)$. These equations are simulated in discrete time using Euler integration.

## 2.2 VARIATIONAL INFERENCE

Note that voltage dynamics in equation 1 is a deterministic equation. We provide further flexibility to the model in order to model the significant trial-to-trial variability we observed in Figure 1c. Such variability could be caused by different initial state of neurons, or by unknown, unmeasured inputs from environment. In order to address this, we allow for the voltage dynamics to deviate from the deterministic dynamics, and introduce a Gaussian prior over these deviations. This results in the following stochastic version of the voltage dynamics,

$$\tau_i\dot{v}_i(t) + v_i(t) = s_i^c(t) + s_i^e(t) + v_i^{rest} + o_i(t) + \sigma_i^v\epsilon_i^v(t), \tag{7}$$

Where $\sigma_i^v$ is the noise amplitude, and the standard normal noise term is $\epsilon_i^v \sim \mathcal{N}(0,1)$. With this approach, we re-formulated the neuronal voltage as a latent variable $\boldsymbol{v}$ with the generative model given by 7 and must perform Bayesian inference to estimate their value. Exact Bayesian inference is intractable for this high-dimensional nonlinear stochastic dynamical system. For this reason, we developed a computationally efficient strategy for inferring an approximate posterior distribution $P(\boldsymbol{v}|\boldsymbol{f})$ over the latent variable $\boldsymbol{v}$, given the measured fluorescence data $\boldsymbol{f}$ using variational Bayes in the framework of the variational autoencoder [15], see derivation in Appendix A.1.

## 2.3 INFERENCE NETWORK

The inference network $Q_\phi(\boldsymbol{v}|\boldsymbol{f}, \boldsymbol{o})$ generates the approximate posterior distribution of the latent variable $\boldsymbol{v}$ (voltage for all neurons at all time points) given the measured fluorescence data $\boldsymbol{f}$ and the sensory inputs $\boldsymbol{o}$. It consists of multiple 1D convolutional layers that transform the input data into a Gaussian distributions $v_i(t) \sim \mathcal{N}(\mu_i(t), \sigma_i(t))$ for each neuron $i = 1, ..., N$ at every time point $t = 0, ..., D$. The approximate posterior distribution $Q_\phi(\boldsymbol{v}|\boldsymbol{f}, \boldsymbol{o})$ is factorized as

$$Q_\phi(\boldsymbol{v}|\boldsymbol{f}, \boldsymbol{o}) = \prod_{t=0,...,D} Q_\phi\left(\boldsymbol{v}(t)|\boldsymbol{f}(0), .., \boldsymbol{f}(D), \boldsymbol{o}(0), .., \boldsymbol{o}(D)\right), \tag{8}$$

$D$ is the length of the time window of data fed into the inference network, and $\phi$ is its parameters.

## 2.4 GENERATIVE MODEL

The generative model outputs the calcium fluorescence trace $\boldsymbol{f}$ and neuronal voltage $\boldsymbol{v}$ given sensory inputs $\boldsymbol{o}$. It can be formulated as

$$P_\theta(\boldsymbol{f}, \boldsymbol{v}|\boldsymbol{o}) = P_\theta(\boldsymbol{f}|\boldsymbol{v}, \boldsymbol{o})P_\theta(\boldsymbol{v}|\boldsymbol{o}), \tag{9}$$

where $\theta$ contains parameters of the generative model. $P_\theta(\boldsymbol{v}|\boldsymbol{o})$ is a biophysically realistic connectome-constrained network with passive point-neuron dynamics. It outputs a prior distribution over voltage $\boldsymbol{v}$ given sensory input $\boldsymbol{o}$. We use a nonlinear sensory mapping $H$ to transform our chemosensory input to neuronal stimulus to the sensory neurons. The stimuli is represented as a set of binary vectors $h_k(t)$, which are 1 given the presence of stimulus $k$ at time $t$ and 0 otherwise. $P_\theta(\boldsymbol{f}|\boldsymbol{v}, \boldsymbol{o})$ is another realistic model that maps the voltage $\boldsymbol{v}$ into reconstructed fluorescence $\boldsymbol{f}$, given sensory inputs $\boldsymbol{o}$. The voltages $\boldsymbol{v}$ are sampled from the approximate posterior distributions generated by the inference network. Details of both distributions are described in Appendix A.2.

| model | | current-based | | | conductance-based | | | | | |
|---|---|---|---|---|---|---|---|---|---|---|
| **weight** | **parameter** | **learned dense** | **learned sparse** | **conne. sparsity** | **learned dense** | **learned sparse** | **learned total count** | **conne. sparsity** | **conne. count**[1] | **conne. count**[2] |
| $W_{ji}^c$ | range | $(-\infty,\infty)$ | $(-\infty,\infty)$ | $(-\infty,\infty)$ | $[0,\infty)$ | $[0,\infty)$ | $[0,\infty)$ | $[0,\infty)$ | $[0,\infty)$ | $[0,\infty)$ |
| | $T_{ji}^c$ | 1 | 1 | 1 | $\mathbf{1}[C_{ji}^c>0]$ | 1 | 1 | $\mathbf{1}[C_{ji}^c>0]$ | $\mathbf{1}[C_{ji}^c>0]$ | $\mathbf{1}[C_{ji}^c>0]$ |
| | $M_{ji}^c$ | ✓ | ✓ | ✓ | ✓ | ✓ | ✓ | ✓ | $C_{ji}^c$ | $C_{ji}^c$ |
| | $\alpha^c$ | 0.01 | 0.01 | 0.01 | 0.01 | 0.01 | 0.01 | 0.002 | ✓ | ✓ |
| $W_{ji}^e$ | range | $[0,\infty)$ | $[0,\infty)$ | $[0,\infty)$ | $[0,\infty)$ | $[0,\infty)$ | $[0,\infty)$ | $[0,\infty)$ | $[0,\infty)$ | $[0,\infty)$ |
| | $T_{ji}^e$ | 1 | 1 | 1 | $\mathbf{1}[C_{ji}^e>0]$ | 1 | 1 | $\mathbf{1}[C_{ji}^e>0]$ | $\mathbf{1}[C_{ji}^e>0]$ | $\mathbf{1}[C_{ji}^e>0]$ |
| | $M_{ji}^e$ | ✓ | ✓ | ✓ | ✓ | ✓ | ✓ | ✓ | $C_{ji}^e$ | $C_{ji}^e$ |
| | $\alpha^e$ | 0.01 | 0.01 | 0.01 | 0.01 | 0.01 | 0.01 | 0.065 | ✓ | ✓ |

Table 1: **Definitions and constraints for the LVM variants.** We evaluated a total of 9 model variants. Across these variants, we explored two methods of modeling chemical synapses: current-based and conductance-based (Sec. 2.1). We also tested several different levels of connectome constraint. (Sec. 2.6). For both the chemical synapse $W_{ji}^c$ and electrical synapses $W_{ji}^e$, the weight matrix $W_{ji} = \alpha T_{ji} M_{ji}$, where $T_{ji}$ is the sparsity matrix, $M_{ji}$ is the matrix of connection magnitudes, and $\alpha$ is a global scalar. Trainable parameters are indicated by a ✓. We compared the connectome-constrained versions to two unconstrained learned models, a learned dense model in which all neurons are connected, a learned sparse model using a $L_1$ regularization $\|\boldsymbol{M}\|_1$, and a learned total count model with a $L_1$ regularization with connectome synapse count $\boldsymbol{C}$ using $\|\|\boldsymbol{M}\|_1 - \|\boldsymbol{C}\|_1\|_2^2$.

## 2.5 Objective function for generative model and inference network

The variational objective of our latent variable model uses the ELBO. The reconstructed calcium fluorescence $\boldsymbol{f}$ is then fit to the measured fluorescence data by maximizing the ELBO for each measured time point. The objective function is a combination of reconstruction loss between the reconstructed and measured fluorescence traces and the KL divergence between voltage prior and posterior. (details in Appendix A.4, A.5):

$$\mathcal{L} = -\mathbb{E}_{\boldsymbol{v}\sim Q(\boldsymbol{v}|\boldsymbol{f},\boldsymbol{o})}\left[\log\left(P_\theta\left(\boldsymbol{f}|\boldsymbol{v},\boldsymbol{o}\right)\right)\right] + D_{\mathrm{KL}}\left(Q_\phi(\boldsymbol{v}|\boldsymbol{f},\boldsymbol{o})\|P_\theta(\boldsymbol{v}|\boldsymbol{o})\right), \tag{10}$$

Optimization of the KL divergence with the reconstruction loss allows our inference network to learn the time dynamics of our neuronal voltages because the difference between the posterior and prior distributions is made small.

## 2.6 Connectome constraint for model weights

To build the model using chemical and electrical synapse count, $\boldsymbol{C}^c$ and $\boldsymbol{C}^e$, as a connectome constraint, we factorized the chemical synaptic weight $W_{ji}^c$ as $W_{ji}^c = \alpha^c T_{ji}^c M_{ji}^c$, where $\alpha^c$ represents a global scaling factor which scales synapse counts to per-synapse currents or conductances depending on the model class. $T_{ji}^c$ is a binary value which indicates the connectivity, and $M_{ji}^c$ reflects the magnitude of connection strength. The electrical synaptic weights matrix $W_{ji}^e$ is similarly defined as $W_{ji}^e = \alpha^e T_{ji}^e M_{ji}^e$. We compared the performance of the connectome constrained models to unconstrained ones (Table 1, Appendix A.6). The anatomical connectome can be represented mathematically as two $N \times N$ matrices: one containing the chemical synapse counts ($\boldsymbol{C}^c$), and one containing the sizes of electrical synapses ($\boldsymbol{C}^e$). We designed the *connectome sparsity* constraint by fixing the sparsity of our model to that of the connectome. For chemical connections, the model sparsity $T_{ji}^c = \mathbf{1}[C_{ji}^c > 0]$, and likewise for electrical connections, $T_{ji}^e = \mathbf{1}[C_{ji}^e > 0]$. For both chemical and electrical connections, the magnitudes $M_{ji}^c$ and $M_{ji}^e$ are trainable.

We applied the connectome sparsity constraint to both current and conductance-based versions of the LVM. In the conductance-based model, we were able[1] to apply the even stronger *connectome count* constraint: directly assigning the synapse counts to the chemical connection magnitudes ($M_{ji}^c = C_{ji}^c$) and the synapse sizes to the electrical connection magnitudes ($M_{ji}^e = C_{ji}^e$). Under this constraint, both the sparsity and magnitude are fixed, and only the global scaling factors $\alpha^c$ and $\alpha^e$ are trainable. Additionally, we evaluated another variant of this constraint, *connectome count*[2]. This constraint uses parameters of *connectome count*[1] to initialize $M_{ji}^c$. $M_{ji}^c$ is then allowed to deviate from $C_{ji}^c$ during re-training. We compared our connectome-constrained models to two unconstrained models. The *learned dense* model used fully connected sparsity matrices $\boldsymbol{T}^c$ and $\boldsymbol{T}^e$, with the magnitudes $\boldsymbol{M}^c$ and $\boldsymbol{M}^e$ as trainable parameters. The *learned sparse* model uses $L_1$ regularization on both magnitude matrices: $\|\boldsymbol{M}^e\|_1$ and $\|\boldsymbol{M}^c\|_1$. The *learned total count* model preserves the total synapse count for

---

[1]Since in the current-based model, the range of $W_{ji}^c$ is $(-\infty,\infty)$, and because the signs of the synaptic connections are unknown, we could not directly apply the *connectome count* constraint to it.

both magnitude matrices: $\|\|M^c\|_1 - \|C^c\|_1\|_2^2$ and $\|\|M^e\|_1 - \|C^e\|_1\|_2^2$. $\alpha^c$ and $\alpha^e$ are 0.002 and 0.065 respectively, which is the same in the *conne.count*[1] model. Thus, it regularizes the sum of elements in $M^c$ and $M^e$ equal to that of $C^c$ and $C^e$.

## 3 EXPERIMENTS

### 3.1 DATASET

We applied the CC-LVM to a calcium imaging dataset in which immobilized, pan-neuronally labeled *C.elegans* were presented with a panel of chemosensory stimuli (2-butanone, 2,3-pentanedione, and NaCl) [32]. The activity of 170 neurons in the head of the animal was measured across 21 individuals. Each recording lasts for 5 min, with an acquisition rate of 4 Hz. In this dataset, every neuron in the worm brain was identified *in vivo* using a deterministic multicolor landmark, allowing whole brain dynamics to be mapped onto the anatomical connectome with single-cell resolution for the first time. The connectome constraints we applied utilized the anatomical connectivity data from [30]. The complete hermaphrodite *C.elegans* was reconstructed using electron microscopy, and the connections between its 300 neurons were mapped. The connectome contains 3464 chemical synapses and 1031 electrical synapses. More details can be found in appendix A.6.

### 3.2 NEURON HOLDOUT EVALUATION

We hypothesized that the CC-LVM which incorporates the complete connectome but has access only to the measured activity of a subset of the neurons may nevertheless be able to accurately predict the activity of unmeasured neurons. This would constitute experimentally testable predictions from our mechanistic model. We tested this hypothesis by performing neuron holdout evaluations, withholding a single bilateral pair of measured neurons from the model during *both* training and testing. Note that our latent variable model always infers the voltages of *all* neurons regardless of how many neurons are observed in any experiment, since the latent states are the voltages of the entire nervous system. The goal of this evaluation is to test the model's ability to predict the neural activity of neurons which were never observed at any time. We fit a version of the model with a pair of neurons withheld, and then compare the model predictions for the same pair of neurons with the ground truth calcium fluorescence measurements. We repeated this evaluation for all 107 measured neuron pairs. Because many of the neurons in *C. elegans* are bilaterally symmetric, the symmetric pairs tend to have high correlations with each other. In order to prevent the model from predicting one neuron solely based on its sibling, we removed the symmetric neurons in groups of two. For every neuron pair, we performed the neuron holdout evaluation for both current and conductance models with different levels of connectome constraints. For these experiments, we calculated the *correlation coefficient* (details in Appendix A.4) between the predicted and measured calcium fluorescence magnitudes to quantify the predictive performance (Figure 2a and Table 2, more predicted traces in Appendix A.9). For each evaluation, we trained 4 models with random initialization, and reported both the mean value and standard error (SE) for each model. We found *connectome sparsity* and *connectome count* constrained models were better at predicting traces of held-out neurons than unconstrained models.

### 3.3 WORM HOLDOUT EVALUATION

Another evaluation method we performed was to hold out the data from a handful of individual worms, train the model on the remaining worms, and predict the activity of the neurons of the held-out individuals. For each of the 9 model variants, we trained on 15 worms, and tested the model on 6 withheld worms. In each case, we trained 4 models with different random initialization. For the worm holdout traces, we calculated the *correlation coefficient* between the predicted and measured calcium fluorescence. We also reported *ELBO*, *reconstruction loss* and *KL divergence* (Appendix A.4) on tested worms to represent the fitting performance. We reported the mean and standard error (SE) from models trained with 4 different random initializations. We found that models using the *connectome count* constraints performed better at predicting calcium traces from held out worms in Table 2.

| model | | | current-based | | | conductance-based | | | | | |
|---|---|---|---|---|---|---|---|---|---|---|---|
| holdout | metric | | learned dense | learned sparse | conne. sparsity | learned dense | learned sparse | learned total count | conne. sparsity | conne. count[1] | conne. count[2] |
| neuron | $corr\uparrow$ | mean | -0.104 | 0.001 | 0.087 | 0.081 | 0.008 | -0.033 | 0.262 | 0.318 | **0.319** |
| | | SE | ±0.013 | ±0.010 | ±0.021 | ±0.012 | ±0.011 | ±0.011 | ±0.016 | ±0.016 | ±**0.016** |
| worm | $corr\uparrow$ | mean | 0.465 | 0.439 | 0.440 | 0.457 | 0.445 | 0.470 | 0.463 | **0.474** | 0.468 |
| | | SE | ±0.006 | ±0.007 | ±0.006 | ±0.006 | ±0.006 | ±0.006 | ±0.006 | ±**0.006** | ±0.006 |
| | $ELBO\uparrow$ | mean | -0.772 | -0.826 | -0.803 | -0.849 | -0.862 | -0.931 | -0.898 | -0.854 | **-0.697** |
| | | SE | ±0.037 | ±0.036 | ±0.036 | ±0.033 | ±0.035 | ±0.033 | ±0.032 | ±0.036 | ±**0.066** |
| | $recon\downarrow$ | mean | 0.745 | 0.765 | 0.754 | 0.771 | 0.780 | 0.805 | 0.082 | 0.744 | **0.636** |
| | | SE | ±0.037 | ±0.037 | ±0.035 | ±0.033 | ±0.035 | ±0.032 | ±0.031 | ±0.035 | ±**0.065** |
| | $KLD\downarrow$ | mean | **0.028** | 0.061 | 0.049 | 0.078 | 0.082 | 0.126 | 0.077 | 0.110 | 0.061 |
| | | SE | ±**0.001** | ±0.003 | ±0.002 | ±0.003 | ±0.002 | ±0.005 | ±0.003 | ±0.006 | ±0.002 |

Table 2: **Holdout evaluation results.** We performed neuron holdout evaluations on each of the 107 measured neuron pairs in the dataset, and report the average *correlation coefficient* between the predicted and measured fluorescence traces (Sec. 3.2). We also performed worm holdout evaluations, witholding 6 of the 21 individuals from the training (Sec. 3.3). For the worm holdouts, we report several metrics: *correlation coefficient*, *ELBO*, *reconstruction loss*, and *KL divergence*. Overall, the conductance-based model under *connectome count* constraint produced the best predictions among the 9 models.

## 3.4 COMPARING THE DIFFERENT MODEL VARIANTS

The gold standard for comparing generative models in machine learning is to compare the model evidence $P(\boldsymbol{f})$ on a test dataset. However, this is intractable for many generative models, and so the evidence lower bound (*ELBO*) is used for model fitting and sometimes also for model comparison. However, the *ELBO* is not a tight lower bound, and so is not guaranteed to provide the same ordering over models as the model evidence $P(\boldsymbol{f})$. In this paper, exploiting the interpretable nature of our mechanistic generative model, we developed a stricter criterion for comparing models based on the neuron holdout evaluation. Neuron holdout predictive performance is biologically meaningful, as it evaluates the ability of a model to make testable predictions about neurons which were not directly measured. We found that overall, the conductance-based CC-LVMs under *connectome count* constraint make the best neuron holdout predictions as quantified by the *correlation coefficient*. We found that model accuracies at neuron holdout prediction are not very tightly correlated with their test *ELBO*. While *connectome count*[2] still achieves the best performance, the *learned dense* model achieves the second best performance, despite performing worst in the neuron holdout evaluation. This suggests that the *learned dense* model has potentially discovered an equally accurate but mechanistically incorrect way to model the calcium imaging data.

**Stronger connectome constraint improves the neuron holdout predictions**. We found that our CC-LVM with the *connectome count* constraint produced significantly better predictions than CC-LVM with the *connectome sparsity* constraint, and both connectome constrained models outperformed unconstrained LVMs with the same dynamics (Table 2). This suggests that the neuron activity predictions are improved by a stronger connectome constraint. Note that *connectome count*[2] achieves slightly better performance than *connectome count*[1], as its parameters get initialized from *connectome count*[1], then are allowed to deviate during re-training. We also explored whether CC-LVMs achieve better performance due solely to the smaller number of trainable parameters. Comparing the connectome-constrained models to the *learned sparse* LVM with sparsity regularization, we found that the learned sparse model performed worse. This indicates that the connectome constraint improves predictability due to its topology, not due to the number of trainable parameters.

Perhaps relatedly, we found superior predictions for interneurons and motor neurons, compared to sensory neurons. While the inputs to motor neurons and especially interneurons are largely contained in the connectome, the inputs to sensory neurons are not as well constrained by the connectome since they also respond to external stimuli. Instead, we trained a black-box model to predict the tunings of sensory neurons to experimentally presented odors, but this likely does not fully characterize all the sensory inputs to the worm. This result additionally suggests the utility of the connectome, for neurons where the connectome provides information about the dominant sources of input.

**Conductance-based models outperform current-based models.** We found that conductance-based models achieve better performance in neuron holdout evaluations, despite needing to estimate the reversal potential $\boldsymbol{E}$ in addition to the non-negative strength of a chemical synapse $\boldsymbol{W}^c$. This could be for several reasons. The biophysical model of a conductance-based synapse might be a more accurate description of reality. There are also two technical advantages to this model. First, conductance-based synapses allow us to make the fullest use of the connectomic data by using the non-negative synapse counts. Second, separating the optimization of the sign of a synapse from its magnitude might allow

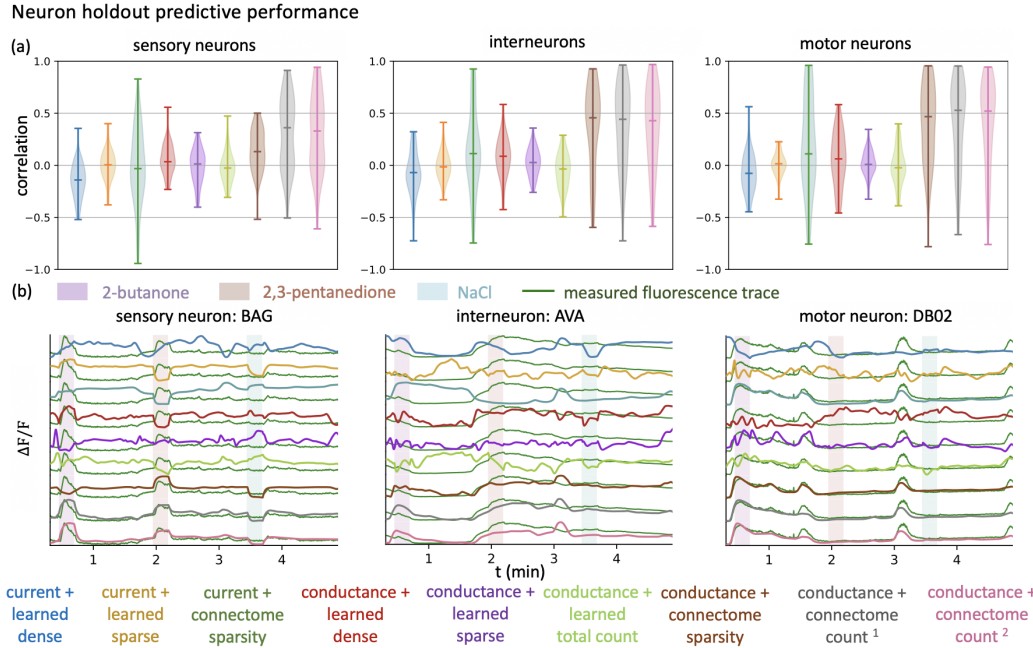

Figure 2: **LVM predictive performance across neuron types. a.** Violin plots comparing the distribution of correlation coefficients between predicted and measured neuron activity for the 9 LVM variants. Neurons are divided into sensory, inter, and motor neurons. **b.** Measured traces and predictions made by the 9 models for three selected neurons: sensory neuron BAG, interneuron AVA, and motor neuron DB02.

for easier optimization of both. In contrast, the sign and magnitude of a current-based synapse are coupled in the same parameter, and we have observed strong local minima preventing the efficient optimization of the sign and the magnitude. Accordingly, we found that conductance-based models converge to a connectome closer to the counts given by the original data with less variance due to the removal of the neuron sign parameter (Appendix A.7). We found that a synapse initialized as excitatory is unlikely to change its sign after optimization. For the same reasons, we are unable to effectively use the non-negative synapse counts contained in the connectomic data to inform the magnitude of the current-based synapse while leaving the sign unconstrained.

## 4 DISCUSSION

In this work, we proposed a connectome-constrained latent variable model (CC-LVM) of the *C.elegans* nervous system. Our latent variable framework enables the estimation of the initial state of the network dynamics and compensates for modeling imperfections. This flexibility allows us to fit models to calcium imaging data with significant trial-to-trial variability. In contrast to black-box generative models used in LVMs in neural data modeling [21], we developed a biophysics-based, connectome-constrained, mechanistically detailed generative model for the neural dynamics. We evaluated current-based and conductance-based models for synapses. We used connectomic data to constrain the model to different degrees. Since CC-LVMs enable the prediction of neurons which were not used to fit the model, we were able to simulate and validate the process of making biologically relevant and experimentally testable predictions of the activity of unmeasured neurons. We found that the conductance-based model produces much more accurate predictions in neuron holdout evaluation and worm holdout evaluation. Knowledge of the connectome strongly constrains *in silico* predictions of individual neurons whose activity were not measured. We hope that this mechanistic model can be used to predict the effects of experimental perturbations, leading to a causal understanding of neural computation in neural circuits. The research in this paper was enabled by recent advances in whole-brain calcium imaging in identified neurons [32].

While we have only analyzed the first datasets to be collected in this manner, we hope that a rich database of such recordings across a diversity of conditions will lead to achieving the dream of standard models of the *C. elegans* nervous system [11] which generalize across different sensory modalities and behaviors. Rapidly developing whole-brain imaging technologies and advances in high-throughput connectomics are now making high-dimensional neural activity and anatomical connectivity data available in other organisms [23; 33; 6; 16]. We hope that this approach will eventually lead to accurate mechanistic models of these larger, more complex biological networks.

## 5 Reproducibility

We trained each of our LVMs on 1 Quadro RTX 8000. Training procedures are described in Appendix A.3, A.4, A.5. We released our software and datasets (`https://github.com/TuragaLab/wormvae`) for reproducibility.

## 6 Acknowledgement

This work was funded by the Howard Hughes Medical Institute. Lu Mi is supported by Intel and MathWorks. Albert Lin and Aravinthan Samuel were funded by NSF Physics of Living Systems (NSF 1806818); NSF Ideas (NSF IOS-1555914); and the NIH (1 U01 NS111697-01). We thank William Bishop, Roman Vaxenburg, Core Francisco Park, Helena Casademunt for comments on the manuscript.

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

# A APPENDIX

## A.1 VARIATIONAL INFERENCE

We used Bayesian inference (VAE) to compute the posterior probability over the latent variables $\boldsymbol{v}$ (neuron voltages) given the data $\boldsymbol{f}$ (measured calcium fluorescence),

$$P(\boldsymbol{v}|\boldsymbol{f}) = \frac{P(\boldsymbol{f}|\boldsymbol{v})P(\boldsymbol{v})}{P(\boldsymbol{f})} \tag{11}$$

To solve this problem, we needed to compute the model evidence,

$$P(\boldsymbol{f}) = \int P(\boldsymbol{f}|\boldsymbol{v})P(\boldsymbol{v})d\boldsymbol{v}. \tag{12}$$

However, computing $P(\boldsymbol{f})$ is a mathematically intractable problem, so we instead used variational inference to approximate the value of $P(\boldsymbol{v}|\boldsymbol{f})$ [15]. Denoting the approximate posterior $Q(\boldsymbol{v}|\boldsymbol{f})$, we can formulate evidence lower bound (ELBO) as

$$\log P(\boldsymbol{f}) \geq \mathcal{L} = \mathbb{E}_{Q(\boldsymbol{v}|\boldsymbol{f})}[\log P(\boldsymbol{f}, \boldsymbol{v}) - \log Q(\boldsymbol{f}|\boldsymbol{v})], \tag{13}$$

which tightens the lower bound on $\log P(\boldsymbol{f})$ as $\mathcal{L}$ is maximized. This allowed us to improve the inference network and find the posterior such that $Q(\boldsymbol{v}|\boldsymbol{f}) \approx P(\boldsymbol{v}|\boldsymbol{f})$, since the Kullback-Leibler (KL) divergence, which measures the distance between the distributions $Q$ and $P$, is minimized by maximizing ELBO.

## A.2 PRIOR AND POSTERIOR DISTRIBUTION AT A DISCRETE TIME STEP

Using Euler discretization with a time step of $\Delta t$, the first-order leaky integrator equation in Equation 1 in the network model is

$$\tau_i(v_i(t + \Delta t) - v_i(t))/\Delta t + v_i(t) = s_i^c(t) + s_i^e(t) + v_i^{rest} + o_i(t + \Delta t), \tag{14}$$

Then the voltage $v_i(t + \Delta t)$ of neuron $i$ at time step $t + \Delta t$ is formulated as

$$v_i(t + \Delta t) = \Delta t/\tau_i(s_i^c(t) + s_i^e(t) + v_i^{rest} + o_i(t + \Delta t) - v_i(t)) + v_i(t), \tag{15}$$

This equation indicates that the voltage state $v_i(t)$ at the current time step $t$ is determined by a function over multiple components from the last time step $t - \Delta t$, including the past voltage state, the chemical and electric synaptic inputs, and the sensory input:

$$v_i(t + \Delta t) = \mathcal{F}(v_i(t), s_i^c(t), s_i^e(t), o_i(t + \Delta t)), \tag{16}$$

For the posterior distribution $Q_\phi(\boldsymbol{v}(t)|\boldsymbol{f}(t), \boldsymbol{o}(t))$, the inference network outputs $\mu_i(t)$ and $\sigma_i(t)$ for every neuron $i$ at each time step $t$. The posterior distribution for the voltage generated by the inference network at each time step is $v_i(t) \sim \mathcal{N}(\mu_i(t), \sigma_i(t))$, given a sequence of measured fluorescence data $\boldsymbol{f}(0), ..., \boldsymbol{f}(D\Delta t)$ and sensory input $\boldsymbol{o}(0), ..., \boldsymbol{o}(D\Delta t)$ with total number of steps $D + 1$. We then used the reparameterization trick to sample the voltage $v_i(t)$ from this posterior distribution to generate a sequence of voltage sample from $v_i(0), ..., v_i(D\Delta t)$.

For the prior distribution, we used the voltage samples $v_i(t)$ from the posterior distribution at the last step as the past voltage state in $\mathcal{F}$ to calculate the prior distribution $P_\theta(v_i(t + \Delta t)|o_i(t + \Delta t))$ at time step $t + \Delta t$. The prior distribution is defined as $\mathcal{N}(\mathcal{F}(v_i(t), s_i^c(t), s_i^e(t), o_i(t + \Delta t)), \sigma_i^v)$. $\sigma_i^v$ is the constant standard deviation of the prior distribution that is trainable during optimization. At the initial time step 0, we defined another trainable parameter $\mu_i(0)$ to represent the mean of the voltage prior distribution.

Similarly, we used Euler integration to discretize the first-order leaky integrator equation (Equation 5). The calcium concentration $[Ca]_i(t + \Delta t)$ of neuron $i$ at time step $t + \Delta t$ is given by

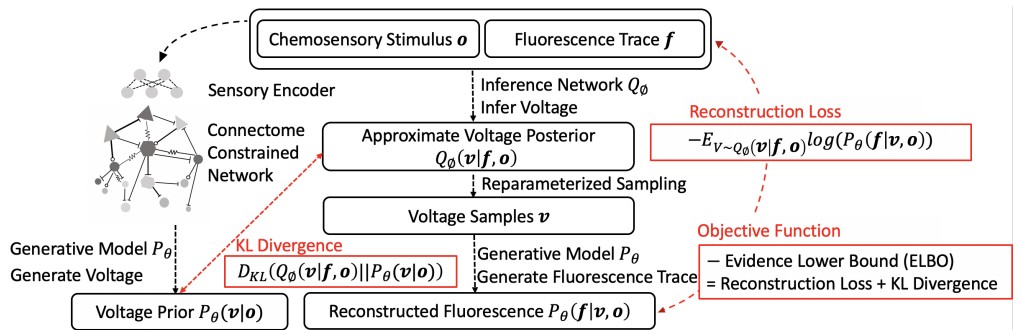

Figure 3: **Schematic of the connectome-constrained latent variable model (CC-LVM).** Sensory inputs are fed through a biophyiscally realistic connectome-constrained network to generate the voltage prior $P_\theta(v|o)$. To train the model, we first used an inference network to infer from the data the approximate voltage posterior $Q_\phi(v|f)$. Fluorescence traces are then reconstructed via a generative model $P_\theta(f|v,o)$. We then compared the reconstructed fluorescence traces to the measured traces, and optimized the model with evidence lower bound (ELBO), with a combination of reconstruction loss and KL divergence.

$$[Ca]_i(t + \Delta t) = \Delta t/\tau_{[Ca]}(g(v_i(t)) - [Ca]_i(t)) + [Ca]_i(t), \tag{17}$$

We used a sequence of voltage samples $v_i(t)$, $t = 0, ..., D\Delta t$ as the inputs to this formula to get a sequence of calcium concentration $[Ca]_i(1), ..., [Ca]_i(D\Delta t)$. The initial concentration is estimated from the measured fluorescence trace at time step 0 using $[Ca]_i(0) = (F_i(0) - \beta^f)/\alpha^f$. Equation 6 in the generative model $P_\theta$ generates the output distributions $P_\theta(f(t)|v(t), o)$ of fluorescence trace at time step $t$ as $\mathcal{N}(\alpha^f[Ca]_i(t) + \beta^f, \sigma^f)$. The values $\alpha^f, \beta^f, \sigma^f$ are trainable fluorescence parameters.

## A.3 MODEL ARCHITECTURE

The calcium fluorescence data is given in a matrix of dimensions $(N, T)$ where N is the total number of neurons in the connectome. The chemosensory stimuli input is modeled as a matrix of dimensions $(3, T)$ which consists of 3 one-hot encoding vectors that are 1 when the stimuli is present and 0 otherwise. Each of the 3 vectors represents one of the three chemical stimuli presented: 2-butanone and 2,3-pentanedione, NaCl.

The calcium fluorescence matrix is passed to the input layer of our inference network $Q(v|f, o)$ which is a $1D$ convolutional layer. The output of the input layer then gets passed through an upsampling layer, a ReLU nonlinearity, another $1D$ convolutional layer, and then another upsampling layer. The first convolutional layer has $2N$ input channels, $2N$ output channels and a kernel size of 11; the second has $2N$ input channels, $2N$ output channels and a kernel size of 21. The first upsampling layer upsamples by a factor of 4 and the second upsamples by a factor of 10. These factors account for the fact that we simulate our network $r$ times faster with upsampling factor $r = 40$ than the recording rate. The sensory input is passed through a $1D$ convolutional layer with $N$ input channels, $2N$ output channels and a kernel size of 21. This is then passed through a ReLU and then concatenated to the output of the 2nd convolution on our calcium data. The merged matrix is then $1D$ convolved again with $2N$ inputs, $2N$ outputs and a kernel size of 41. This is one of the final outputs of our inference network and gives a $(N, T)$ matrix that represents the means of our posterior distribution. To generate the standard deviation of our factorized posterior distribution, we pass our merged data through another convolutional layer with the same dimensions but different parameters. We pass that output through a nonlinear affine transform (using a softplus nonlinearity) to rescale the output to an appropriate level of noise. The parameters of this inference network are all the weights and biases of the convolutions and upsampling layers described above, which we denote by $\phi$ in the paper.

The input to the generative model is the same sensory input that we pass to our inference network. The first layer of our generative model is a single layer nonlinear transform that converts our 3-dimensional input to an $N$ dimensional input that is 0 to any non-sensory neurons. The next layer is our recurrent neural dynamics, given by Equation 1, which passes information to itself in the next time step based on the connectivity given by the connectome. Although some sources say that *C. elegans* has 302 neurons, we only simulate 300 because two cells originally classified as neurons are not connected to the rest of the connectome Witvliet et al. [31]. The simulated voltage for each neuron in the connectome is passed to our calcium simulation layer (Equation 5). The final layer of

our model is calcium to fluorescence affine transformation (Equation 6). The output of this final layer is then compared to the ground truth fluorescence data to calculate the reconstruction loss. Below, we have included some pseudocode describing the architecture of our network.

```
Inference_Network(calcium_fluor, missing_data_mask, sensory_input):
        calcium_input = concatenate(calcium_fluor, missing_data_mask)
        conv1_out = relu(conv1(calcium_input))
        up1_out = upsample(conv1_out)
        conv2_out = relu(conv2(up1_out))
        up2_out = upsample(conv2_out)

        sensory_conv_out = relu(conv3(sensory_input))
        merged_calcium_sensory = concatenate(up2_out, sensory_conv_out)

        mean_latent_neuron_voltage = conv4(merged_calc_sensory)

        std_latent_neuron_voltage = softplus(conv5(merged_calcim_sensory))

        sample_latent_neuron_voltage = mean_latent_neuron_voltage
        + rand_norm * std_latent_neuron_voltage

        return sample_latent_neuron_voltage

Generative_Model(sample_latent_neuron_voltage, sensory_input):

        #Equation 1
        neuron_voltage_dynamics = leaky_integrator_connectome_dynamics(
        sample_latent_neuron_voltage, sensory_input)

        #Equation 5
        calcium_concentration_dynamics = leaky_integrator_calcium_model(
        neuron_voltage_dynamics)

        #Equation 6
        fluorescence_trace = nonlinear_affine_transform(
        calcium_concentration_dynamcis)

        return fluorescence_trace
```

## A.4    FORMULATIONS FOR OBJECTIVE FUNCTIONS AND METRICS

Here we provide the mathematical formulations for the objective functions and metrics we used in this work.

The evidence lower bound (ELBO) is a combination of reconstruction log likelihood and the negative KL divergence. The reconstruction log-likelihood, given the predicted fluorescence trace $\hat{f}_i(t)$ of neuron $i$ at time step $t$ as output distribution $\mathcal{N}(\hat{f}_i(t), \sigma^f)$. And $\hat{f}_i$ is formulated as

$$\hat{f}_i(t) = \alpha^f [Ca](t) + \beta^f. \tag{18}$$

Given the measured fluorescence trace $f_i(t)$, number of neurons $N$, and the total number of time steps $D + 1$, we calculated the reconstruction negative log likelihood as the reconstruction loss

$$\mathcal{L}_{\text{recon}} = -\sum_{t=0}^{D\Delta t} (2\pi(\sigma^f)^2)^{-N/2} \exp\left(-\frac{1}{2(\sigma^f)^2} \sum_{i=1}^{N} (f_i(t) - \hat{f}_i(t))^2\right) \tag{19}$$

The KL divergence describes the distance between voltage prior and posterior distribution. We represent the posterior distribution generated from the inference network as $\mathcal{N}(\mu_i(t), \sigma_i(t))$ for

neuron $i$ at each time step $t$. We then sample $v_i(t)$ from the posterior distribution. According to Equation 16, we represent the mean $\hat{v}_i(t)$ of prior distribution $\mathcal{N}(\hat{v}_i(t), \sigma_i^v)$ as

$$\hat{v}_i(t) = \mathcal{F}(v_i(t), s_i^c(t), s_i^e(t), o_i(t + \Delta t)), \tag{20}$$

where $\sigma_i^v$ is a trainable parameter.

We then use a closed-form of KL divergence defined between two univariate Gaussian distributions

$$D_{\mathrm{KL}} = \sum_{t=0}^{D\Delta t} \sum_{i=1}^{N} \left( \log\left( \frac{\sigma_i^v}{\sigma_i(t)} \right) + \frac{(\sigma_i(t))^2 + (\mu_i(t) - \hat{v}_i(t))^2}{2(\sigma_i^v)^2} - \frac{1}{2} \right) \tag{21}$$

We minimize the objective function $\mathcal{L} = -ELBO$ during optimization given by

$$\mathcal{L} = \mathcal{L}_{\mathrm{recon}} + D_{\mathrm{KL}} \tag{22}$$

In the worm holdout evaluation, for each neuron pair, we calculated the *ELBO*, *reconstruction loss* and *KL divergence* metrics on each holdout worm, for a total 6 worms. We also trained each model setting 4 times under the same constraints and different initializations. We calculated the mean and standard error of *ELBO*, *reconstruction loss* and *KL divergence* based on $4 \times 6 = 24$ samples for each model.

For both neuron and worm holdout evaluations, we calculated the correlation coefficients between the measured $f_i(t)$ and reconstructed fluorescence traces $\hat{f}_i(t)$. The correlation coefficient for each neuron is simply

$$r_i = \frac{\displaystyle\sum_{t=0}^{D\Delta t} \left( \hat{f}_i(t) - \bar{\hat{f}}_i(t) \right) \left( f_i(t) - \bar{f}_i(t) \right)}{\sqrt{\displaystyle\sum_{t=0}^{D\Delta t} \left( \hat{f}_i(t) - \bar{\hat{f}}_i(t) \right)^2 \sum_{t=0}^{D\Delta t} \left( f_i(t) - \bar{f}_i(t) \right)^2}}. \tag{23}$$

In the neuron holdout evaluation, for each neuron pair, we trained a model in which the ground truth of said neuron pair was removed from the training data. There were 107 pairs and 170 measured neurons in total. We trained each model 4 times with different initializations under each of 8 connectome constraints, for a total of $8 \times 107 \times 4 = 3424$ different models. We calculated the mean and standard error of $r_i$ based on $4 \times 170 = 680$ samples for each model.

In the worm holdout evaluation, we calculated these metrics on each holdout worm, for a total of 6 worms. Each worm has about 170 measured neurons. We trained each model 4 times under the same connectomic constraints and with different initializations. We calculated the mean and standard error of $r_i$ based on $6 \times 170 \times 4 = 4080$ samples for each model in our evaluation.

### A.5 GRADIENT-BASED OPTIMIZATION

We used PyTorch to minimize the ELBO using gradient descent with respect to the parameters of both the generative and inference models,

For LVMs which used *connectome count* constraint with $\alpha^c$ and $\alpha^e$ trainable. The objective function is

$$\mathcal{L} = \mathcal{L}(\tau, \tau_{[\mathrm{Ca}]}, \alpha^c, \alpha^e, \alpha^f, \beta^f, \sigma^v, \sigma^f, \boldsymbol{E}, \mathrm{MLP}, \phi) \tag{24}$$

For LVMs which used other constraints including *connectome count* and *learned dense*, *learned sparse*, where $\boldsymbol{M}^c$ and $\boldsymbol{M}^e$ trainable. The objective function is

$$\mathcal{L} = \mathcal{L}(\tau, \tau_{[\mathrm{Ca}]}, \boldsymbol{M}^c, \boldsymbol{M}^e, \alpha^f, \beta^f, \sigma^v, \sigma^f, \boldsymbol{E}, \mathrm{MLP}, \phi) \tag{25}$$

where $\phi$ is all the parameters of the inference network. We used a clamp in PyTorch to constrain the weight magnitude $\boldsymbol{M}^c$ and $\boldsymbol{M}^e$ to be non-negative in several models. We restrict the $\boldsymbol{M}^e$ to be symmetric by defining it as the sum of an upper triangular matrix and its transpose. Our simulation time step is 6.25ms. Each window size $D$ lasts for 7.5s with 30 imaging steps. We used weights of 1 for both the reconstruction loss and KL divergence. We used Adam with a learning rate scheduler to perform the optimization. We used an initial learning rate of $3e^{-4}$, with a learning rate schedular with a step size of 50, and a gamma of 0.5. We also set a gradient clip value of 1. Each model was trained for 300 epochs in which each epoch is one full pass through all the training data.

In the paper, we use the variable $\theta$ to denote the trainable parameters of our generative model. The parameters in $\theta$ are given by

$$\theta = \{\tau, \tau_{[Ca]}, \boldsymbol{M}^c, \boldsymbol{M}^e, \alpha^f, \beta^f, \sigma^v, \sigma^f, \boldsymbol{E}, H\} \tag{26}$$

where $\tau$ and $\tau_{[Ca]}$ are the time scales of the neuron and calcium dynamics. $\boldsymbol{M}^c$ and $\boldsymbol{M}^e$ are the magnitudes of neuronal connections for chemical and electrical synapses respectively. $\alpha^f$ and $\beta^f$ are the scaling and bias parameters of the fluorescence affine transform. $\sigma^v$ and $\sigma^f$ are the noise magnitudes in the neural dynamics equation (Equation 1) and the fluorescence equation (Equation 6). The variable $H$ represents the parameters of the sensory mapping that transforms chemosensory data into neural stimulus.

The variable $\phi$ denotes all the trainable parameters of the inference network. These are just all the weights and biases of the 5 different convolutional layers described in section A.3.

### A.6 CONNECTOME CONSTRAINTS FOR MODEL WEIGHTS

We constructed CC-LVMs on both current-based and conductance-based synapses with different levels of connectomic constraints. Given anatomical connectome data with two $N \times N$ matrices: chemical synapse counts ($\boldsymbol{C}^c$), and sizes of electrical synapses ($\boldsymbol{C}^e$). The chemical synaptic weight $W_{ji}^c$ as $W_{ji}^c = \alpha^c T_{ji}^c M_{ji}^c$, where $\alpha^c$ represents a global scaling factor. $T_{ji}^c$ is a binary value which indicates the connectivity, and $M_{ji}^c$ reflects the magnitude of connection strength. The electrical synaptic weights matrix $W_{ji}^e$ is similarly defined as $W_{ji}^e = \alpha^e T_{ji}^e M_{ji}^e$. We provide more implementation details of each constraint below.

**conductance-based + *connectome count*[1]** We directly assign the synapse counts to the chemical connection magnitudes ($M_{ji}^c = C_{ji}^c$) and the synapse sizes to the eletrical connection magnitudes ($M_{ji}^e = C_{ji}^e$). Under this constraint, both the sparsity $T_{ji}^c$ and $T_{ji}^e$ are fixed to the connectome sparsity, magnitude $M_{ji}^c$ and $M_{ji}^e$ are fixed to the connectome synapse count, and only the global scaling factors $\alpha^c$ and $\alpha^e$ are trainable. $M_{ji}^c$ and $M_{ji}^e$ are restricted to be non-negative.

**conductance-based + *connectome count*[2]** We initialize all model parameters with the parameters from the trained *conductance-based + connectome count*[1] model. And then the sparsity $T_{ji}^c$ and $T_{ji}^e$ are fixed to the connectome sparsity, global scaling factors $\alpha^c$ and $\alpha^e$ are fixed to 0.01. And now magnitude $M_{ji}^c$ and $M_{ji}^e$ are trainable during the re-training. $M_{ji}^c$ and $M_{ji}^e$ are restricted to be non-negative.

**conductance-based + *connectome sparsity*** For chemical and electrical connections, the model sparsities are $T_{ji}^c = \mathbf{1}[C_{ji}^c > 0]$ and $T_{ji}^e = \mathbf{1}[C_{ji}^e > 0]$ respectively. For both chemical and electrical connections, the magnitudes $M_{ji}^c$ and $M_{ji}^e$ are trainable. The sparsity $T_{ji}^c$ and $T_{ji}^e$ are fixed to the connectome sparsity and global scaling factors, $\alpha^c$ and $\alpha^e$, are fixed to 0.01. $M_{ji}^c$ and $M_{ji}^e$ are restricted to be non-negative.

**conductance-based + *learned total count*** For chemical and electrical connections, the model is fully connected with the sparsity $T_{ji}^c = 1$ and $T_{ji}^e = 1$ respectively. For both chemical and electrical connections, the magnitudes $M_{ji}^c$ and $M_{ji}^e$ are trainable. The model is constrained to preserve the same total synapse count as connectome count $\|\boldsymbol{C}^c\|$ for both magnitude matrices adding $\|\|\boldsymbol{M}^c\|_1 - \|\boldsymbol{C}^c\|_1\|_2^2$ and $\|\|\boldsymbol{M}^e\|_1 - \|\boldsymbol{C}^e\|_1\|_2^2$ in the objective function. $\alpha^c$ and $\alpha^e$ are defined as 0.002 and 0.065, which are the same as the trained *conne.count*[1] model, so that regularizes the sum of elements in $\boldsymbol{M}^c$ equal to that of $\boldsymbol{C}^c$, and likewise for $\boldsymbol{M}^e$ and $\boldsymbol{C}^e$. The sparsity $T_{ji}^c$ and $T_{ji}^e$ are fixed to 1 and $M_{ji}^c$ and $M_{ji}^e$ are restricted to be non-negative.

| model | | current-based | | | conductance-based | | | | | |
|---|---|---|---|---|---|---|---|---|---|---|
| consistency | weight | learned dense | learned sparse | conne. sparsity | learned dense | learned sparse | learned total count | conne. sparsity | conne. count[1] | conne. count[2] |
| inter | $\boldsymbol{W}^c$ | 0.142 | 0.084 | 0.497 | 0.082 | 0.084 | 0.092 | **1.000** | **1.000** | **1.000** |
| | $\boldsymbol{W}^e$ | 0.024 | 0.000 | 0.748 | 0.097 | 0.000 | 0.010 | 0.997 | **1.000** | 0.997 |
| connectome | $\boldsymbol{W}^c$ | -0.002 | -0.001 | 0.469 | -0.001 | 0.001 | 0.003 | **1.000** | **1.000** | **1.000** |
| | $\boldsymbol{W}^e$ | -0.002 | 0.000 | 0.654 | 0.006 | 0.000 | 0.007 | 0.998 | **1.000** | 0.996 |

Table 3: **Inter-consistency and connectome-consistency**. Correlation between multiple model weights with different initializations, and correlation coefficient between model weights and connectome counts. Evaluations are performed for models fitted on a single worm and across current-based and conductance-based models with different levels of connectome constraints. The CC-LVM is more consistent among different random initializations and more highly correlated to the connectome counts.

**conductance-based + *learned sparse*** For chemical and electrical connections, the model is fully connected with the sparsity $T_{ji}^c = 1$ and $T_{ji}^e = 1$ respectively. For both chemical and electrical connections, the magnitudes $M_{ji}^c$ and $M_{ji}^e$ are trainable. Meanwhile, it uses an additional $L_1$ regularization on both magnitude matrices: $\|\boldsymbol{M}^c\|_1$ and $\|\boldsymbol{M}^e\|_1$, so that the fraction of nonzero elements in $\boldsymbol{M}^c$ is small, and likewise for $\boldsymbol{M}^e$. The sparsity $T_{ji}^c$ and $T_{ji}^e$ are fixed to 1 and global scaling factors $\alpha^c$ and $\alpha^e$ are fixed to 0.01. $M_{ji}^c$ and $M_{ji}^e$ are restricted to be non-negative.

**conductance-based + *learned dense*** For chemical connections and electrical connections, the model is a fully connected with sparsity $T_{ji}^c = 1$ and $T_{ji}^e = 1$ respectively. For both chemical and electrical connections, the magnitudes $M_{ji}^c$ and $M_{ji}^e$ are trainable. The sparsity matrices, $T_{ji}^c$ and $T_{ji}^e$ are fixed. Global scaling factors $\alpha^c$ and $\alpha^e$ are also fixed to 0.01. $M_{ji}^c$ and $M_{ji}^e$ are restricted to be non-negative.

**current-based + *connectome sparsity*** For chemical and electrical connections, the model sparsity is $T_{ji}^c = \mathbf{1}[C_{ji}^c > 0]$ and $T_{ji}^e = \mathbf{1}[C_{ji}^e > 0]$ respectively. For both chemical and electrical connections, the magnitudes $M_{ji}^c$ and $M_{ji}^e$ are trainable. The sparsity $T_{ji}^c$ and $T_{ji}^e$ are fixed to the connectome sparsity and global scaling factors, $\alpha^c$ and $\alpha^e$, are fixed to 0.01. $M_{ji}^e$ are restricted to be non-negative, while the sign of $M_{ji}^c$ is not restricted.

**current-based + *learned sparse*** For chemical and electrical connections, the model is fully connected with sparsity $T_{ji}^c = 1$ and $T_{ji}^e = 1$. For both chemical and electrical connections, the magnitudes $M_{ji}^c$ and $M_{ji}^e$ are trainable. For both chemical and electrical connections, the magnitudes $M_{ji}^c$ and $M_{ji}^e$ are trainable. Meanwhile, it uses an additional $L_1$ regularization on both magnitude matrices: $\|\boldsymbol{M}^c\|_1$ and $\|\boldsymbol{M}^e\|_1$, so that the fraction of nonzero elements in $\boldsymbol{M}^c$ is small, and likewise for $\boldsymbol{M}^e$. The sparsity $T_{ji}^c$ and $T_{ji}^e$ are fixed to 1, global scaling factors, $\alpha^c$ and $\alpha^e$, are fixed to 0.01. $M_{ji}^e$ are restricted to be non-negative, while the sign of $M_{ji}^c$ is not restricted.

**current-based + *learned dense*** For chemical and electrical connections, the model is fully connected with $T_{ji}^c = 1$, and $T_{ji}^e = 1$. For both chemical and electrical connections, the magnitudes $M_{ji}^c$ and $M_{ji}^e$ are trainable. The sparsity $T_{ji}^c$ and $T_{ji}^e$ are fixed to 1 and global scaling factors $\alpha^c$ and $\alpha^e$ are fixed to 0.01. $M_{ji}^e$ are restricted to be non-negative, while the sign of $M_{ji}^c$ is not restricted.

## A.7   Model weight consistency analysis

We used the *correlation coefficient* to quantify the inter-consistency of models trained with different random initializations, as well as the connectome-consistency, which is the correlation coefficient between model weights and connectome counts.

In particular, for the connectome consistency, we compare the consistency of our model weights and connectome synapse count. We calculate the correlation coefficient $r_i$ between the absolute value of model weights $|W_{ij}|$ and connectome synapse count $C_{ij}$ along each row $i$. Then it is averaged across $N$ rows and 4 models with identical constraints but different random initializations to get $\bar{r}_i$ as our metric for connectome-consistency,

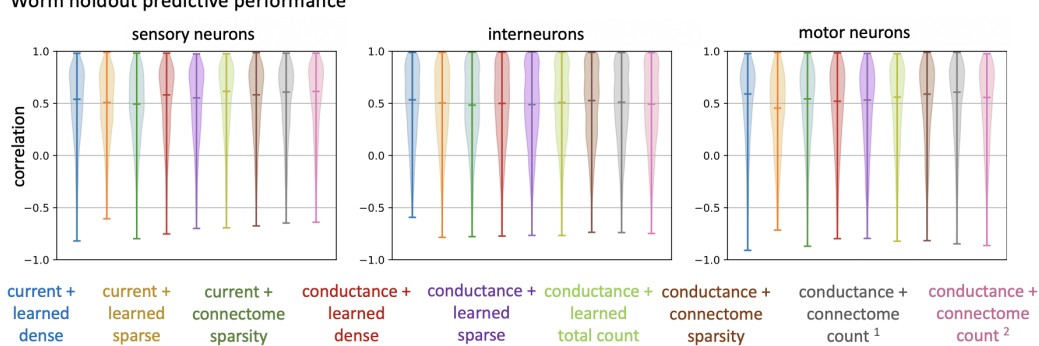

Figure 4: **Worm holdout evaluation**: Violin plots comparing the distribution of correlation coefficients between predicted and measured neuron activity for the 9 LVM variants. Neurons are divided into sensory, inter, and motor neurons.

$$r_i = \frac{\sum_{j=1}^{N} \left( |W_{ij}| - |\bar{W}_{ij}| \right) \left( C_{ij} - \bar{C}_{ij} \right)}{\sqrt{\sum_{j=1}^{N} \left( |W_{ij}| - |\bar{W}_{ij}| \right)^2 \sum_{j=1}^{N} \left( C_{ij} - \bar{C}_{ij} \right)^2}}. \tag{27}$$

For inter-consistency, we evaluate how model weights are affected by random initialization after training. We calculate the correlation coefficient $r_i$ between the weights $W_{ij}^1$ of model 1 and weights $W_{ij}^2$ of model 2 along each row $i$. Model 1 and model 2 have the same constraints but different random initializations. The correlation coefficient between them is

$$r_i = \frac{\sum_{j=1}^{N} \left( W_{ij}^1 - \bar{W}_{ij}^1 \right) \left( W_{ij}^2 - \bar{W}_{ij}^2 \right)}{\sqrt{\sum_{j=1}^{N} \left( W_{ij}^1 - \bar{W}_{ij}^1 \right)^2 \sum_{j=1}^{N} \left( W_{ij}^2 - \bar{W}_{ij}^2 \right)^2}}. \tag{28}$$

This coefficient, $r_i$, is then averaged across $N$ rows and each combination of 2 models within the 4 models with random initialization to get $\bar{r}_i$ as our metric for inter-consistency.

As shown in Table 3, the conductance-based model parameters are more consistent under different random initializations. Optimization of the conductance-based model is easier because the chemical weights $\boldsymbol{W}^c$ of the conductance-based synapses are always positive and we do not need a trainable sign variable. Meanwhile, the model parameters are better constrained with the connectome sparsity or counts, which enables it to be consistent across different initialization.

### A.8 PREDICTIVE PERFORMANCE FOR WORM HOLDOUT V.S. NEURON HOLDOUT

We demonstrate the violin plot for the model performance in worm holdout evaluation in Figure 4. We found that the results from worm holdout are different from those in the neuron holdout experiment in Figure 2. Here, all 9 LVM variants perform similarly across different categories. Our connectome-constrained models do not outperform other unconstrained baselines.

We also studied whether the predictive performance of worm holdouts and neuron holdouts were correlated. In Figure 5, we show the scatter plot of predictive performance for worm holdouts v.s. neuron holdouts. Each point represents one neuron from the same worm. We also report the correlation coefficient between the predictive performance for worm holdout and neuron holdout for each LVM variant. We found that our connectome-constrained model achieves the highest correlation,

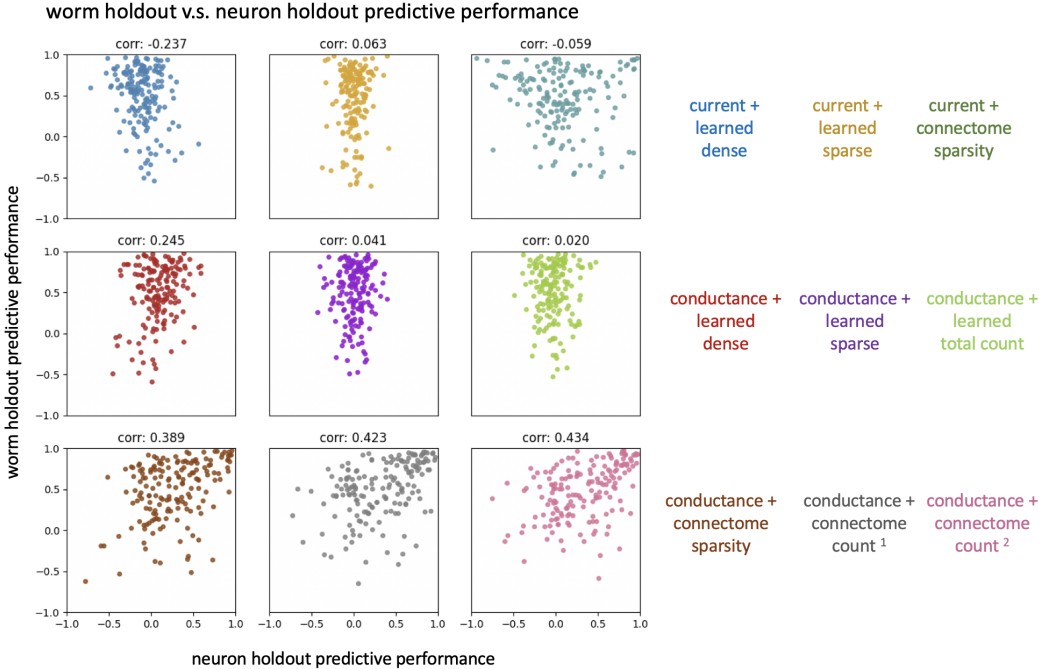

Figure 5: **Scatter plot for predictive performance of worm holdout v.s. neuron holdout.** Each point represents one neuron. Our best model connectome count[2] achieves highest correlation, which indicates highest consistency for neurons which gets better performance in both evaluations.

and also indicates the highest consistency for neuron sets which gets better performance in both evaluations.

## A.9 PREDICTED HOLDOUT TRACES WITH VARIOUS CONSTRAINTS

For each neuron, we performed both neuron holdout evaluation and worm holdout evaluation for current-based and conductance-based models with different levels of connectomic constraint.

For neuron holdout evaluation, we plot additional predicted traces from 15 representative neurons, including 12 with good performance (top), and 3 with worse performance (bottom) in Figure 6, models with connectome count constraint outperforms other baselines.

For worm holdout evaluation, we show the results on the same 15 neurons from 9 LVMs in Figure 7, all models achieve comparable results.

## A.10 PREDICTED TRACES AND VOLTAGE LATENT SPACE FOR NEURON HOLDOUT EVALUATION

For neuron holdout evaluation, we also show the predicted traces and voltage latent space for all measured neurons. We show 189 neurons in Figure 8, Figure 9 and Figure 10. In each subplot, we reported the measured fluorescence trace ground truth, predicted fluorescence trace ground truth, and the mean and standard deviation (std) of voltage latent space (posterior distribution) from the inference network.

## A.11 PREDICTED TRACES AND VOLTAGE LATENT SPACE FOR WORM HOLDOUT EVALUATION

For worm holdout evaluation, similarly, we also show the predicted traces and voltage latent space for all neurons. We show 300 neurons from a train worm in Figure 11–Figure 15. Meanwhile, we also show 300 neurons from a test worm in Figure 16–Figure 20. In each subplot, we reported the measured fluorescence trace ground truth (for recorded neurons), predicted fluorescence trace ground

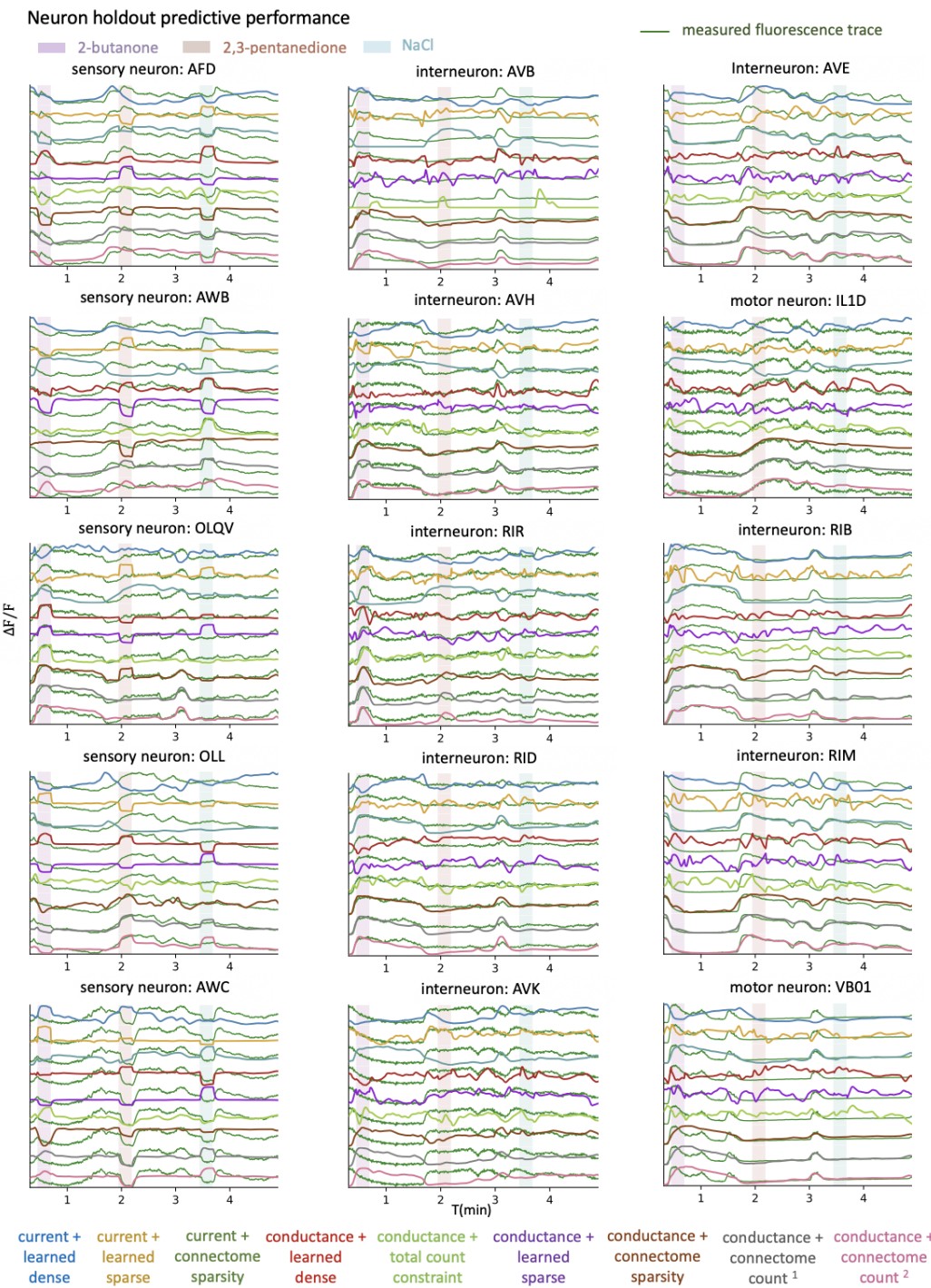

Figure 6: **LVM predictive performance across neuron types for neuron holdout evaluation.** Measured traces and predictions made by the 9 LVMs for 15 neurons including sensory neurons, interneurons and motor neurons.

truth, and the mean and standard deviation (std) of voltage latent space (posterior distribution) from the inference network.

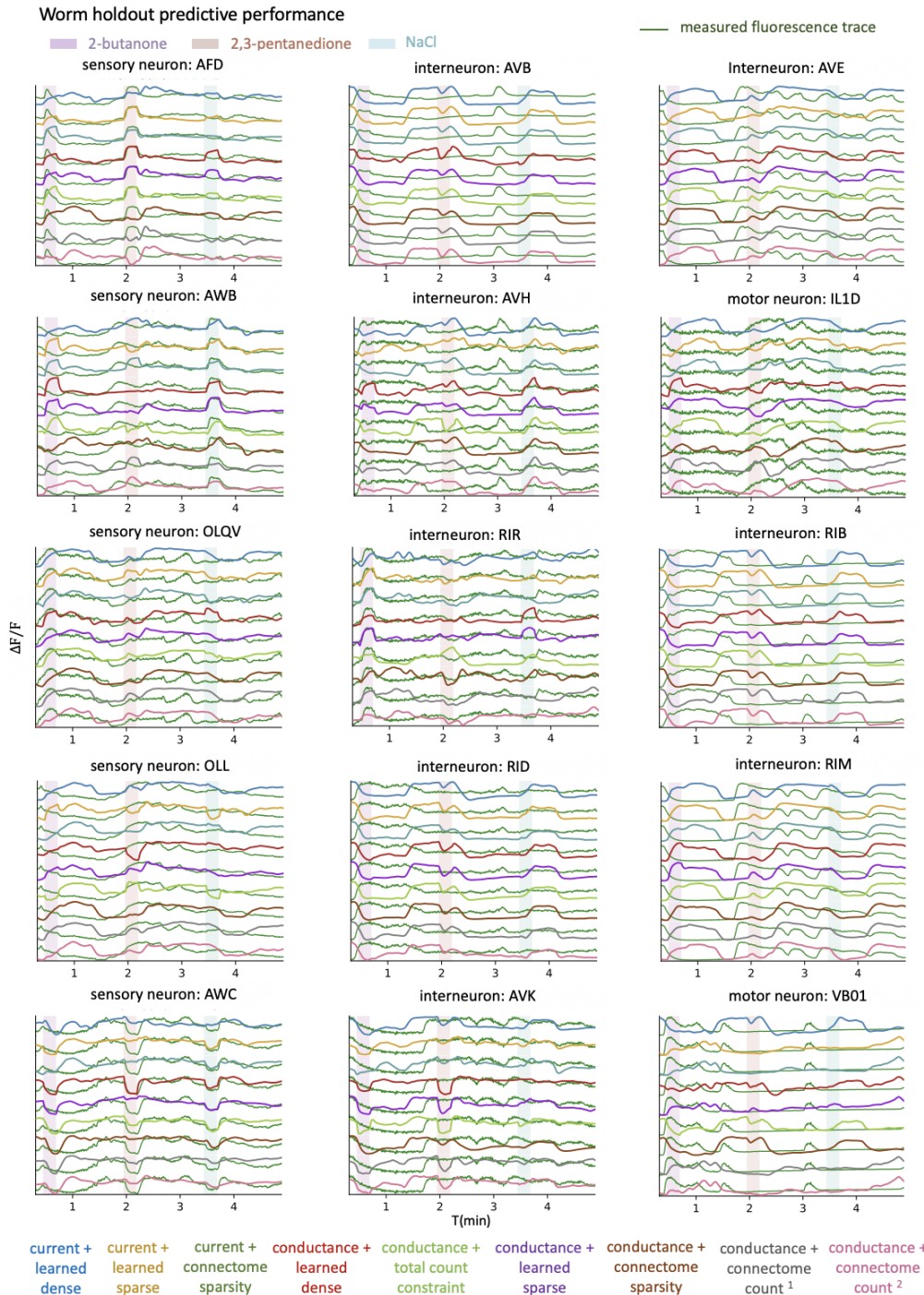

Figure 7: **LVM predictive performance across neuron types for worm holdout evaluation.** Measured traces and predictions made by the 9 LVMs for 15 neurons including sensory neurons, interneurons and motor neurons.

For the train worm, we find most of the neurons have good predictive performance compared to the measured ground truth. That indicates optimization is saturated on the train set. However, the test worm has a relatively worse predictive performance, which indicates the generalization gaps between

each individual trial (animal). While some neurons still achieve good results, that indicates our model is still generalized to unseen individuals to some degree.

Interestingly, for both train and test worms in the worm holdout evaluation, we find those unmeasured neurons still have reasonable patterns. That indicates our model successfully avoids the degeneracy of solutions for those unmeasured neurons.

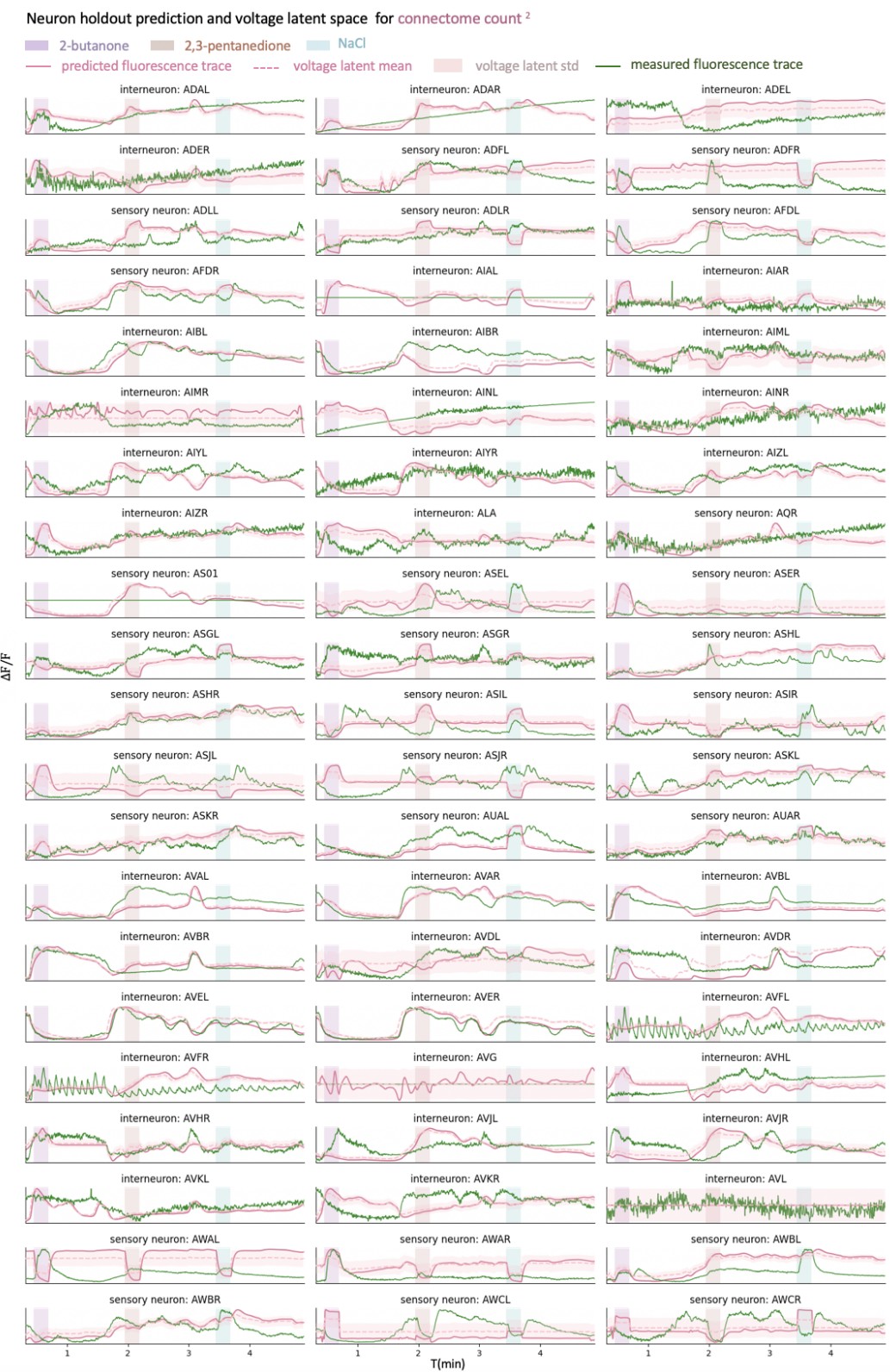

Figure 8: **Neuron holdout evaluated on connectome count**[2]: CC-LVM predicted traces and the corresponding voltage latent space across all recorded neurons (part 1).

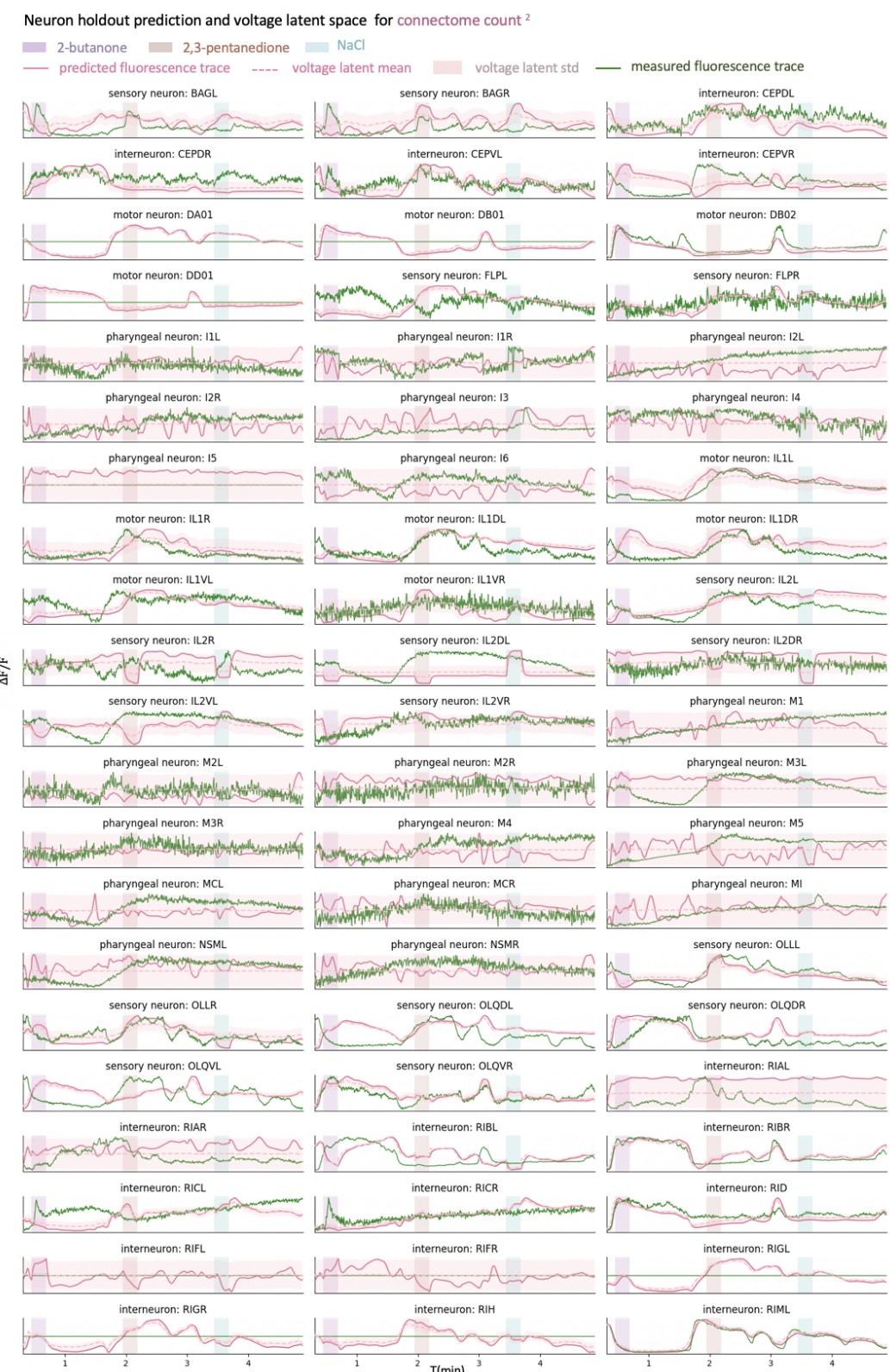

Figure 9: **Neuron holdout evaluated on connectome count**[2]: CC-LVM predicted traces and the corresponding voltage latent space across all recorded neurons (part 2).

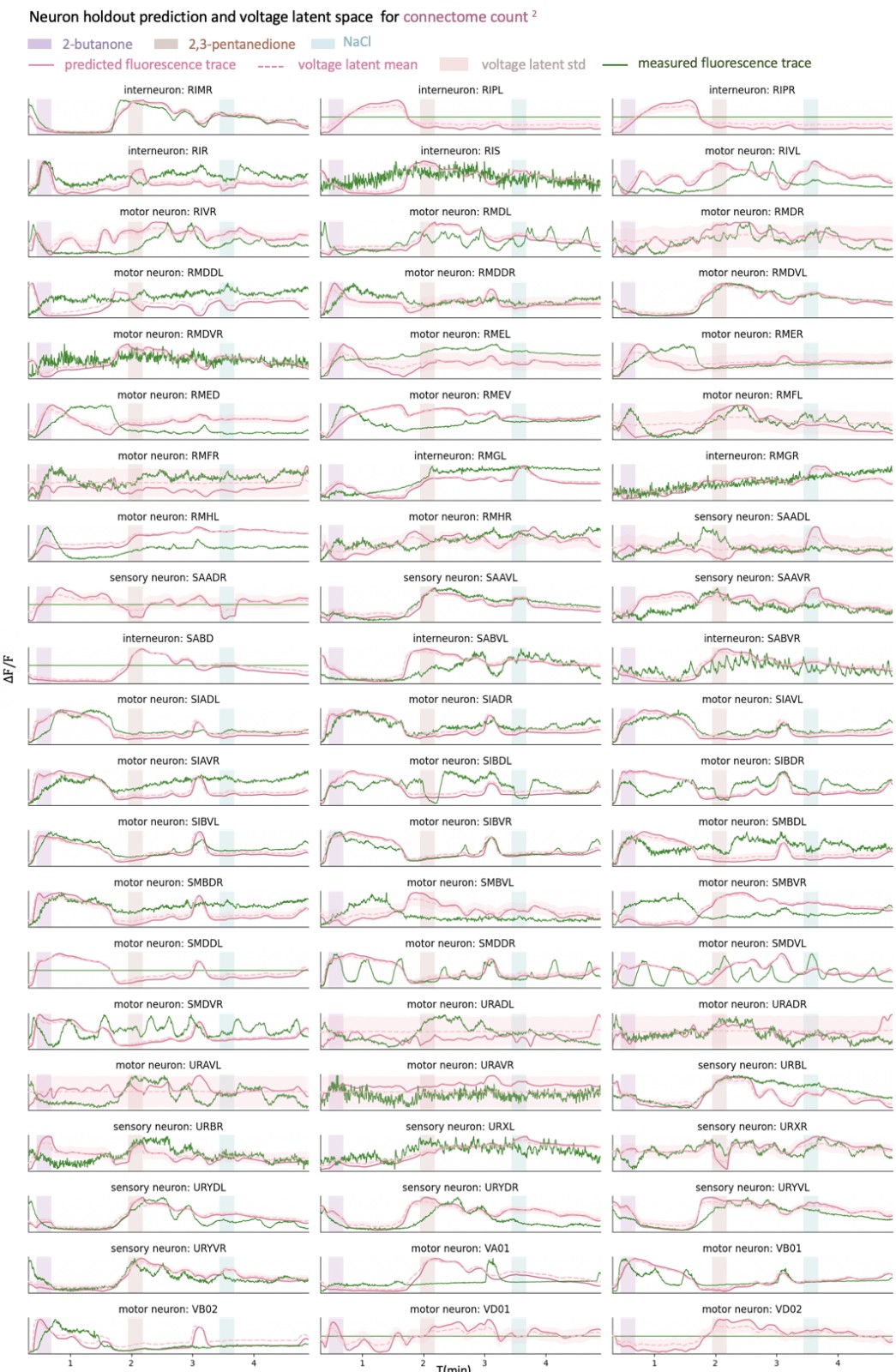

Figure 10: **Neuron holdout evaluated on connectome count**[2]: CC-LVM predicted traces and the corresponding voltage latent space across all recorded neurons (part 3).

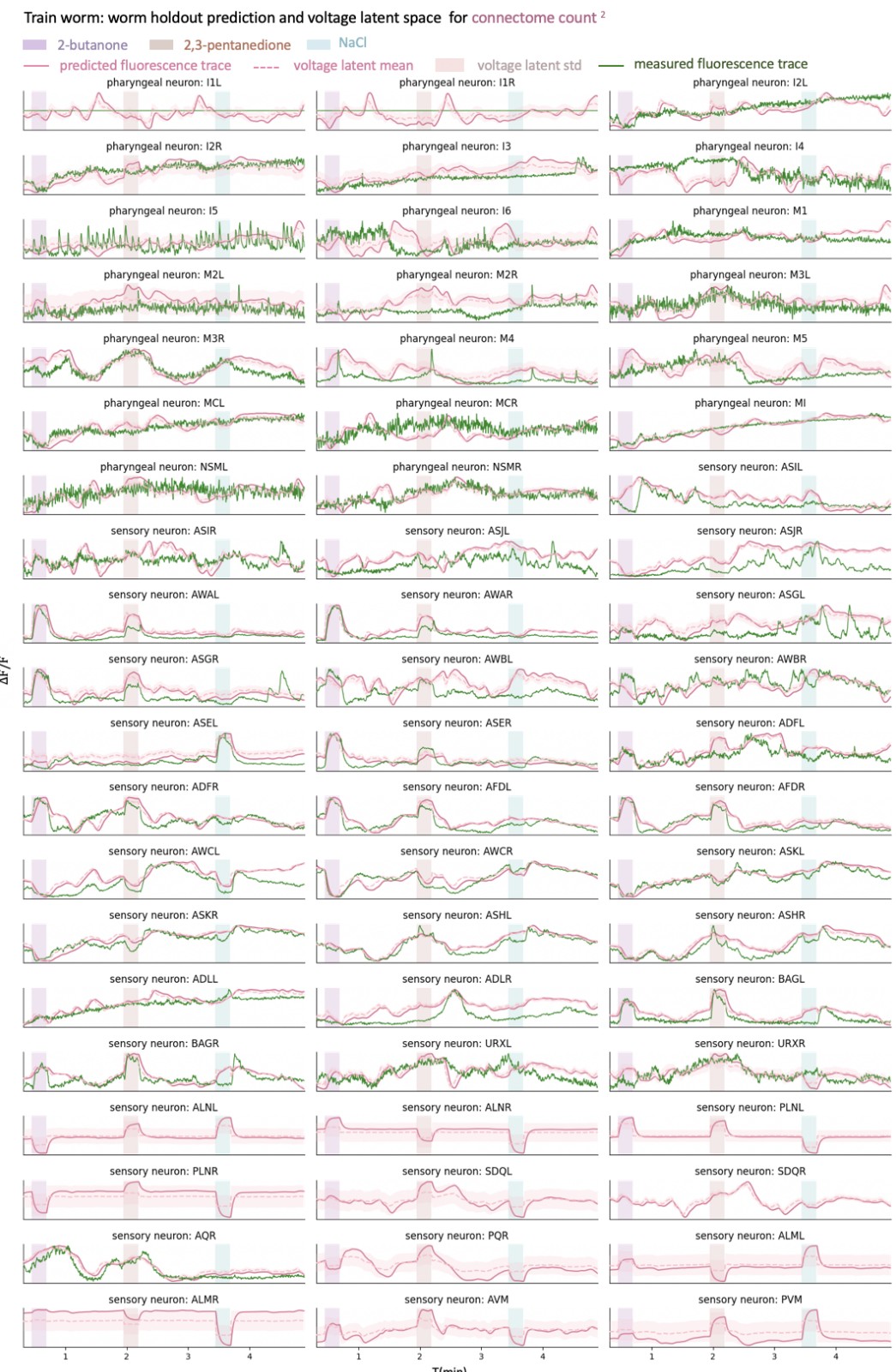

Figure 11: **A train worm evaluated on worm holdout using connectome count$^2$ constraint**: CC-LVM predicted traces and the corresponding voltage latent space across all neurons including unmeasured neurons (part 1).

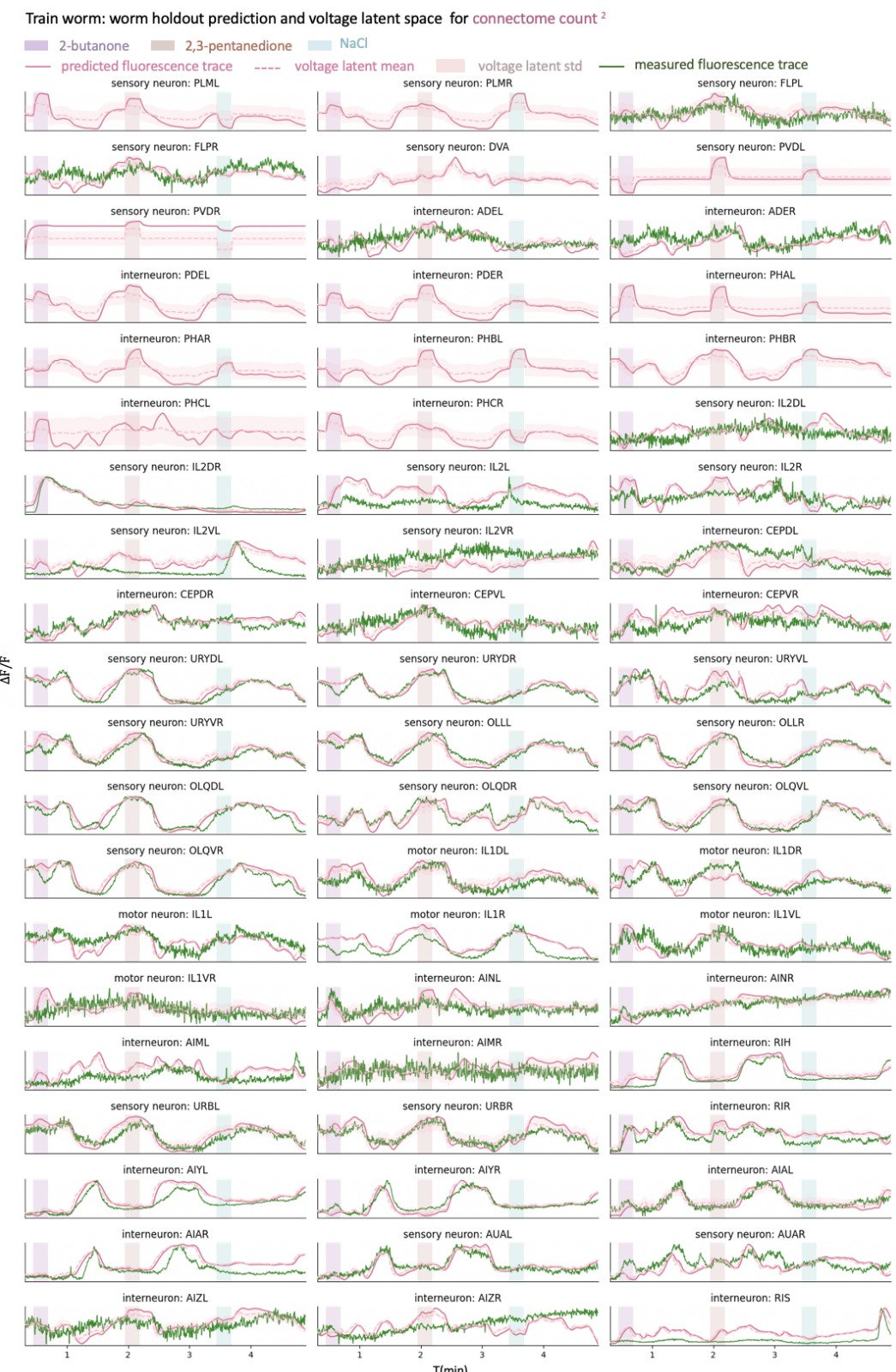

Figure 12: **A train worm evaluated on worm holdout using connectome count$^2$ constraint**: CC-LVM predicted traces and the corresponding voltage latent space across all neurons including unmeasured neurons (part 2).

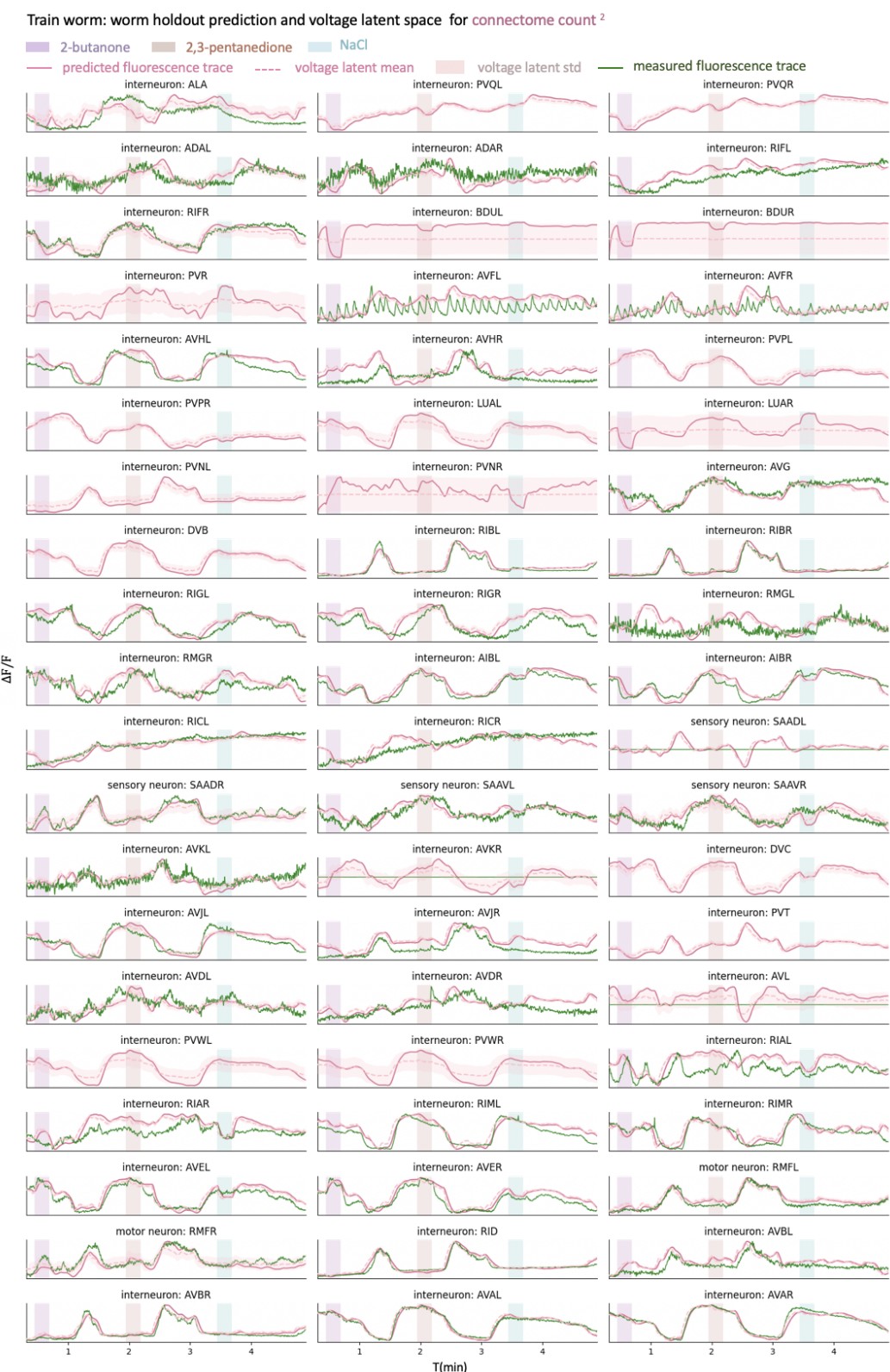

Figure 13: **A train worm evaluated on worm holdout using connectome count$^2$ constraint**: CC-LVM predicted traces and the corresponding voltage latent space across all neurons including unmeasured neurons (part 3).

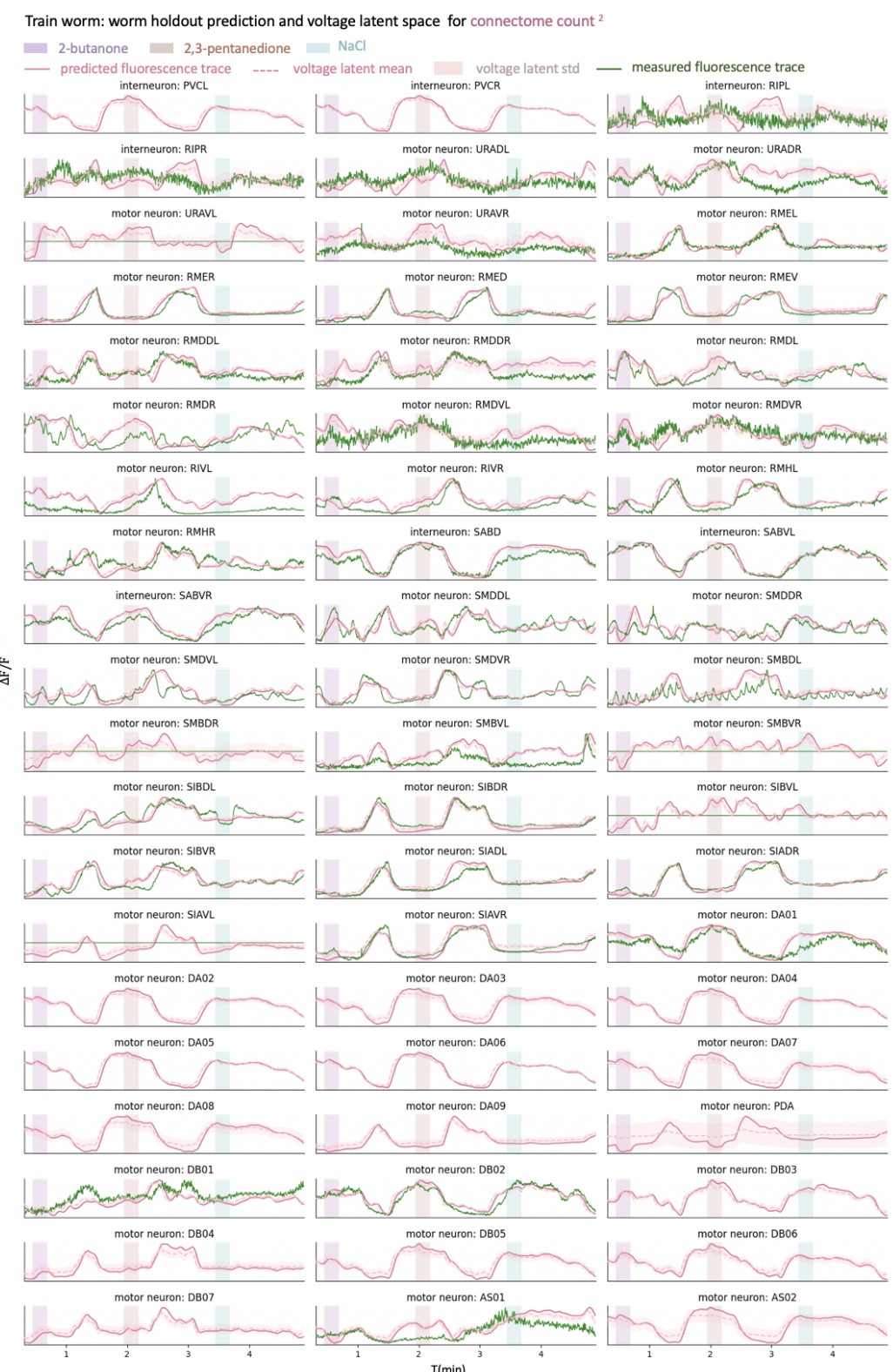

Figure 14: **A train worm evaluated on worm holdout using connectome count$^2$ constraint**: CC-LVM predicted traces and the corresponding voltage latent space across all neurons including unmeasured neurons (part 4).

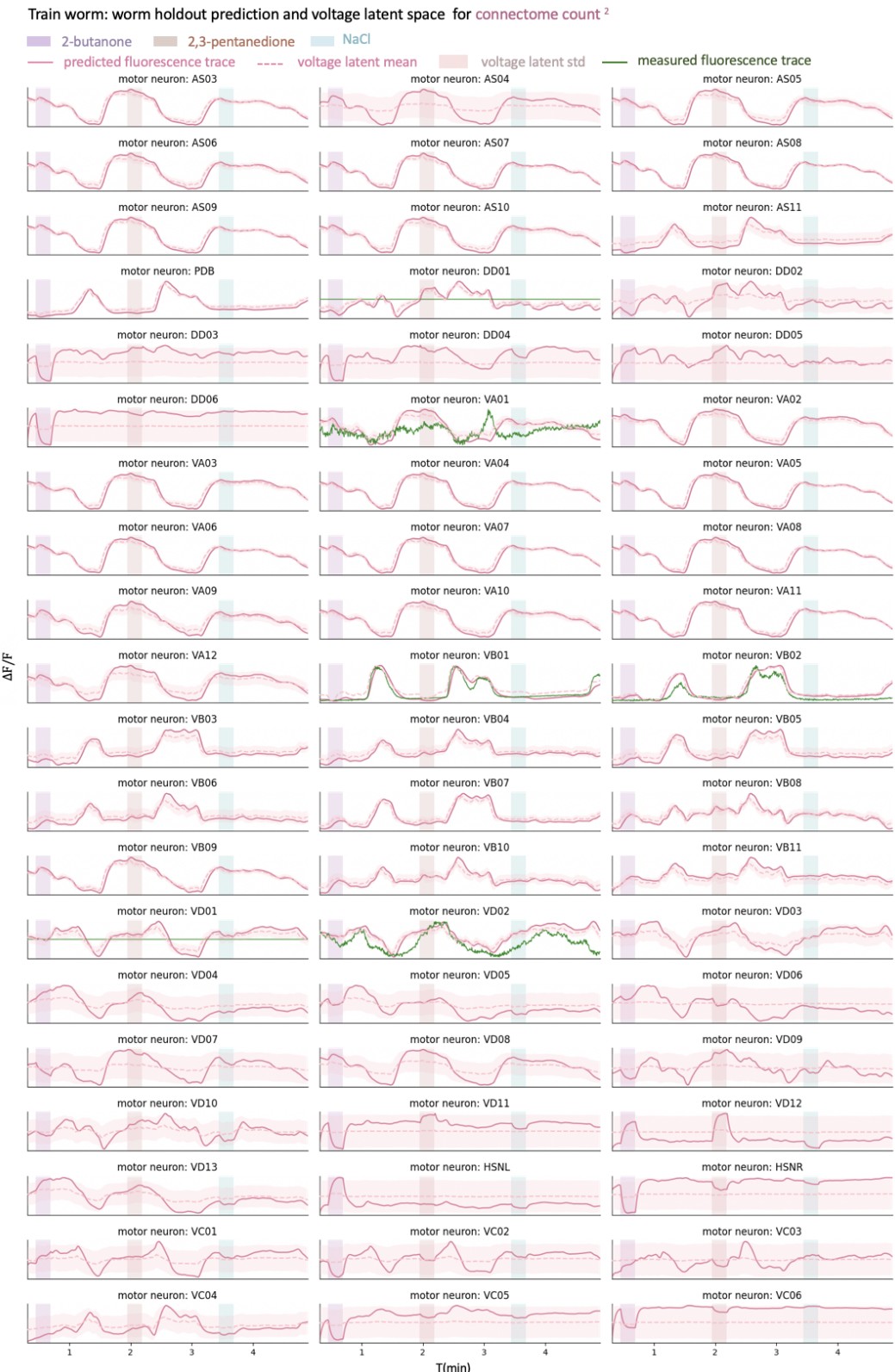

Figure 15: **A train worm evaluated on worm holdout using connectome count$^2$ constraint**: CC-LVM predicted traces and the corresponding voltage latent space across all neurons including unmeasured neurons (part 5).

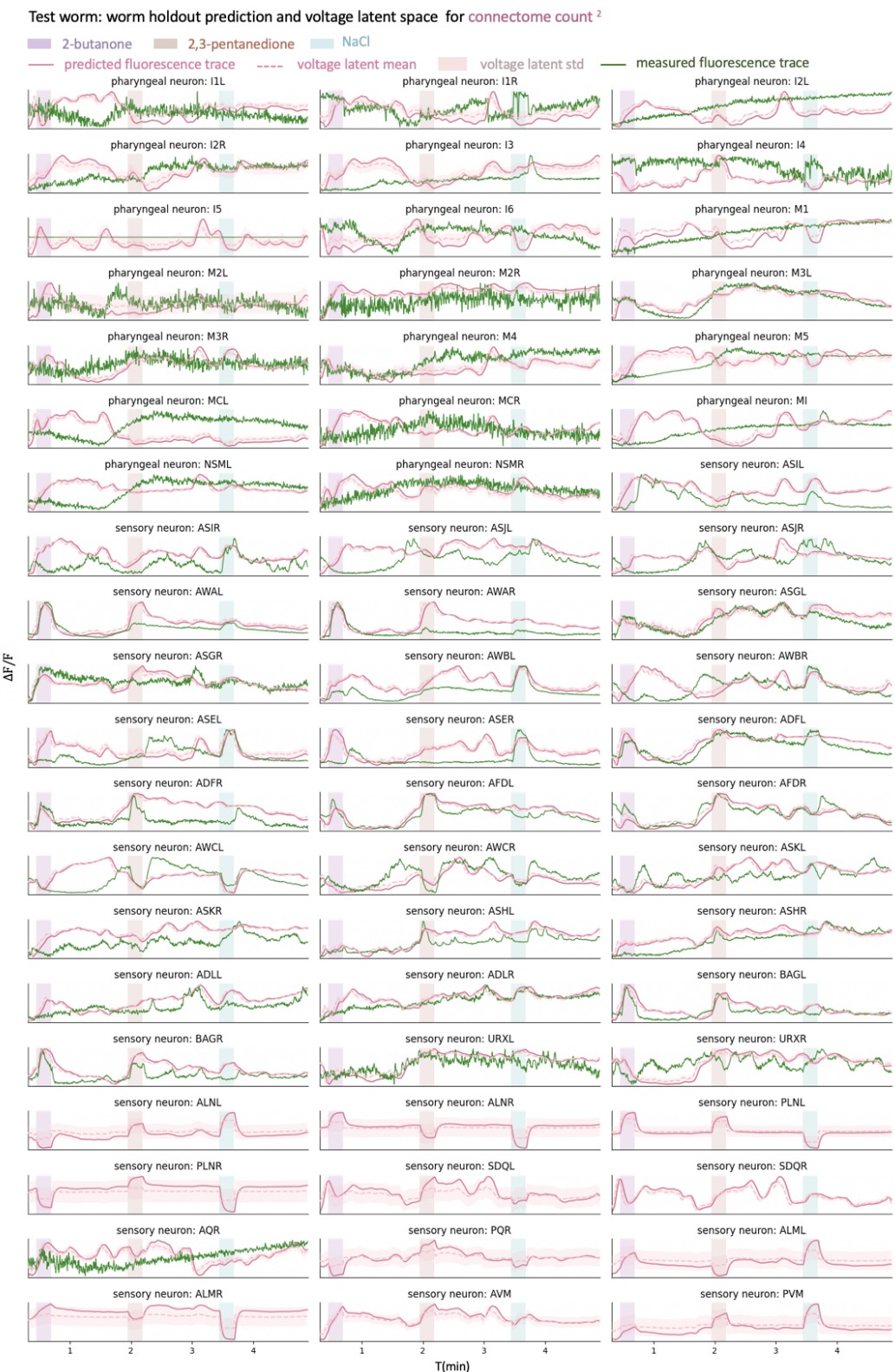

Figure 16: **A hold-out (test) worm evaluated on worm holdout using connectome count$^2$ constraint**: CC-LVM predicted traces and the corresponding voltage latent space across all neurons including unmeasured neurons (part 1).

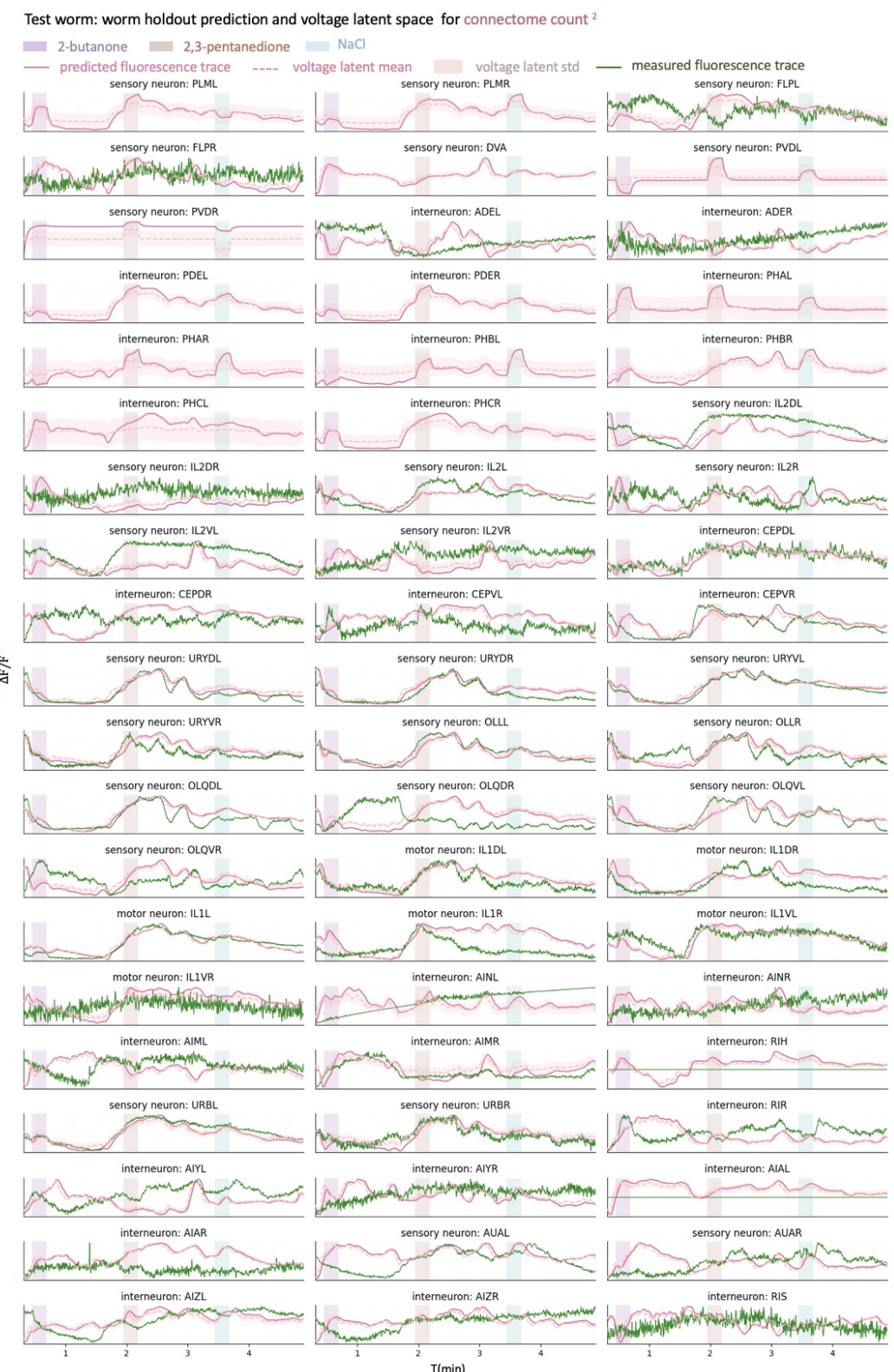

Figure 17: **A hold-out (test) worm evaluated on worm holdout using connectome count$^2$ constraint**: CC-LVM predicted traces and the corresponding voltage latent space across all neurons including unmeasured neurons (part 2).

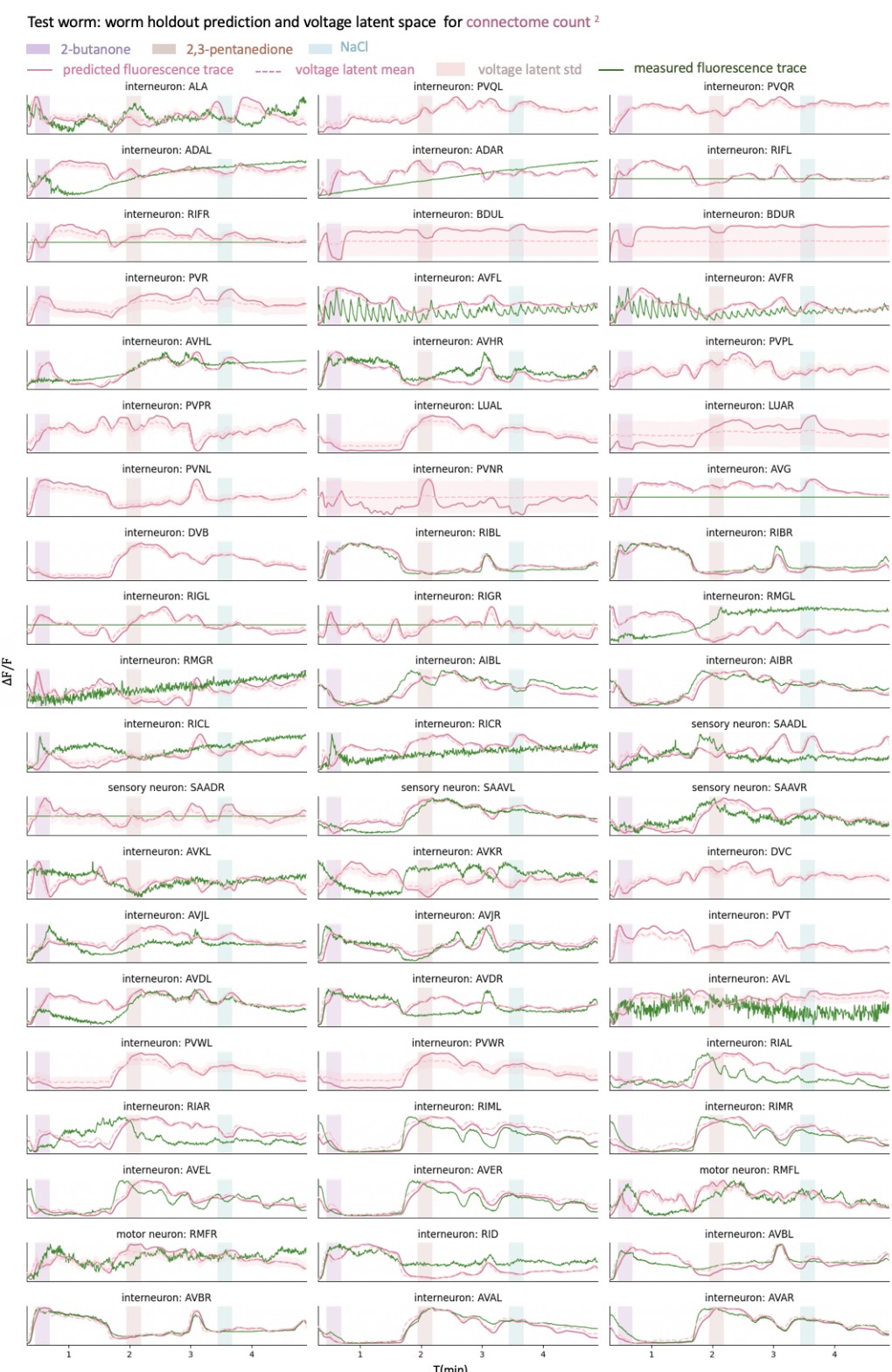

Figure 18: **A hold-out (test) worm evaluated on worm holdout using connectome count**[2] **constraint**: CC-LVM predicted traces and the corresponding voltage latent space across all neurons including unmeasured neurons (part 3).

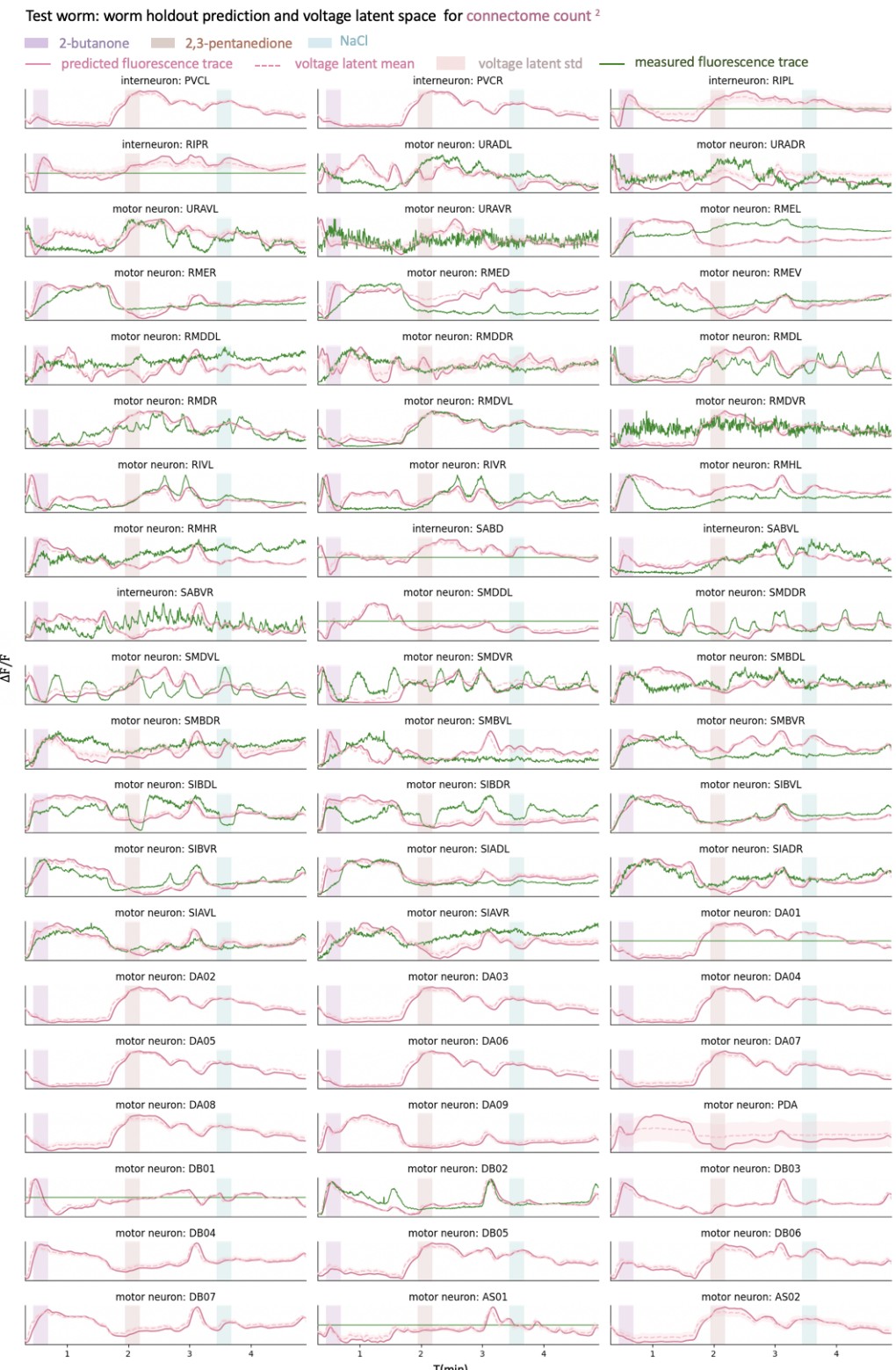

Figure 19: **A hold-out (test) worm evaluated on worm holdout using connectome count$^2$ constraint**: CC-LVM predicted traces and the corresponding voltage latent space across all neurons including unmeasured neurons (part 4).

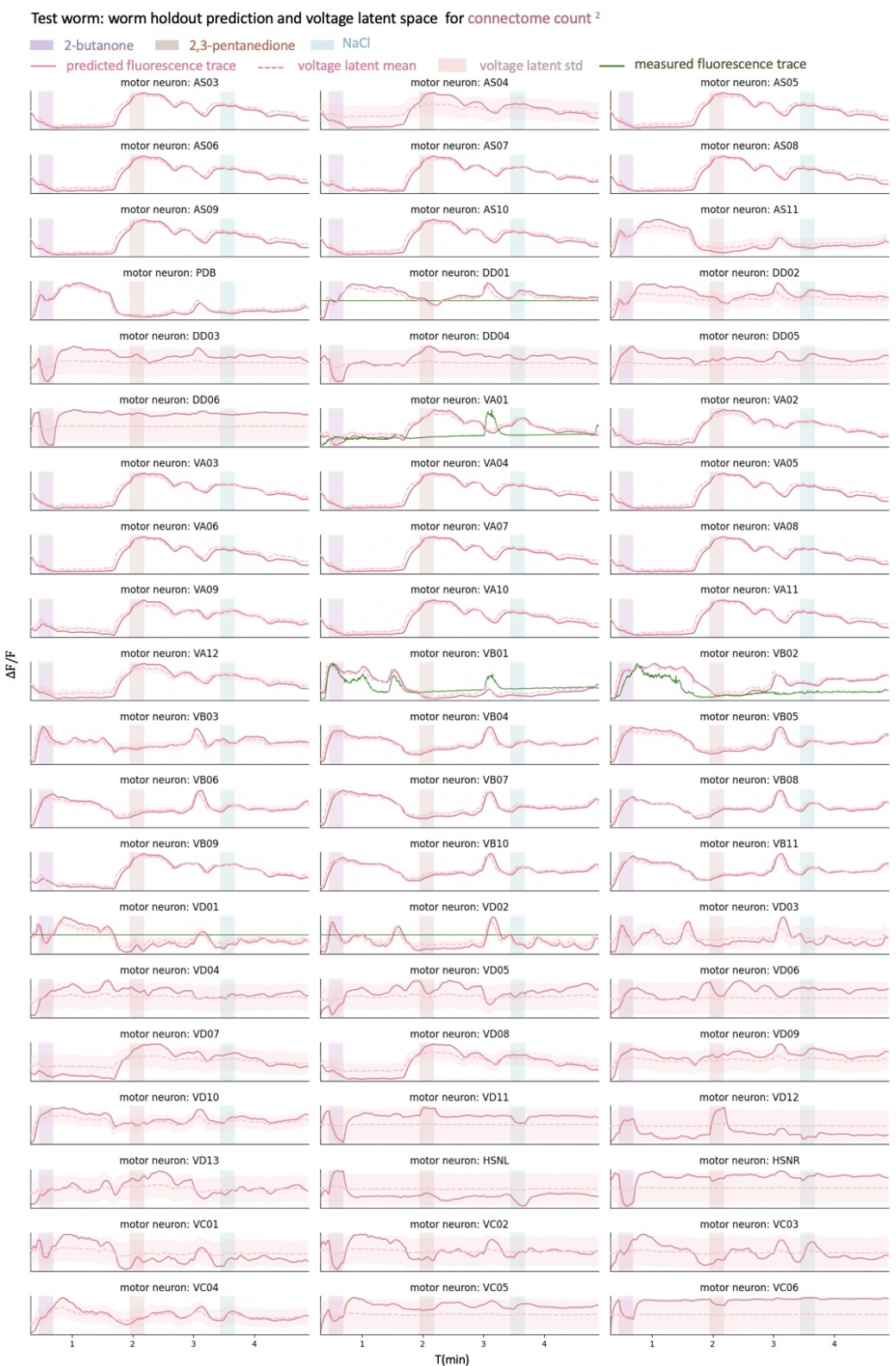

Figure 20: **A hold-out (test) worm evaluated on worm holdout using connectome count$^2$ constraint**: CC-LVM predicted traces and the corresponding voltage latent space across all neurons including unmeasured neurons (part 5).

