# OpenReview forum: "Connectome-constrained Latent Variable Model of Whole-Brain Neural Activity"
_ICLR.cc/2022/Conference — ICLR 2022 Poster_

### Official Review · Reviewer_aiwW · 2021-10-31

**Correctness:** 3
**Technical Novelty And Significance:** 2
**Empirical Novelty And Significance:** 1
**Recommendation:** 3
**Confidence:** 4

**Main Review:**

Strength
- Adding biophysical constraints into neuron activity modeling, and conducting both neuron and worm hold-out experiments to validate the model.
- Alleviate the issue of trail-to-trail variability.

Weakness — not clearly written at all:
- About the prior p(v|h) generation: it only mentions “Pθ(v|h) is a biophysically realistic connectome-constrained network with passive point-neuron dynamics” without pointing out what the network is: are the only trainable parameters in the network W^c_ji and W^e_ji, and a MLP  layer for encoding h_i(t), or are there any other layers? And since it has subscript θ, the same as the decoder network, should I assume they are the same network? If so, how can the decoder be trained since it needs the prior to calculate KL divergence for the loss?
- There’s no detail about the exact network architecture, layer, input/output dimensions, data processing/ training procedure, etc. This makes the whole logic even harder to comprehend.
- Apart from clarity, the technical novelty in computer science is a bit insignificant. Empirically, the results are reasonable but not interesting enough, and no comparison is made with other baseline modeling methods. I would suggest a re-writing (rephrasing, better connecting between sections, putting many appendix contents like A2 and A5 into the main text, etc.) and re-submitting to a neuroscience journal.

**Summary Of The Paper:**

This paper uses VAE (variational autoencoder) to model neural activities of C.elegans observed in calcium imaging. The main contribution is making the encoder-learned posterior distribution close to a biophysically constrained prior, as well as designing the decoder based on biophysical rules.

**Summary Of The Review:**

Overall, the paper provides a good aspect in modeling neuron activities by using connectome constrains as VAE prior, but it’s not clearly written and lacks novelty or significance in both technical and empirical aspects.

---

> ### Author Response · Authors · 2021-11-23
> **Thank you for taking the time to review our paper. Answers related to prior generation, model details and significance.**
>
> Thank you for taking the time to review our paper. Based on your review, we have revised our paper to make it clearer and add significant additional details. Below, we respond to a few of your comments.
>
> **Clarification for $P_{\theta}(v|h)$**
>
> The trainable parameters are not only $W_{ij}^c$ and $W_{ij}^e$, but also include the parameters of our MLP sensory mapping $h(t)$, $\tau$, $v^\text{rest}$, $\tau_\text{[Ca]}$, $\beta_f$ and $\alpha_f$ . The sensory map $h(t)$ is not the encoder of our VAE but rather a part of the generative model that converts chemosensory input into neuronal stimulus for our sensory neurons. We use the variable $\theta$ to denote all the parameters of our generative model, so it includes the parameters of our sensory mapping $h(t)$, as well as the parameters of the connectome-based neural voltage simulator and the voltage to calcium fluorescence model. Accordingly, $\theta$ parameterizes our prior, $P_{\theta }(v|h)$, and $\phi$ parameterizes our posterior which is calculated by our inference network that we denote $Q_{\phi}(v|f,h)$. The KL divergence computes the difference between the two distributions $P$ and $Q$. We realize that the notation we used to describe our parameters can be improved and have provided more information at the end of Appendix A.5 in our revision.
>
> **Model details**
>
> This information has been added to our appendix under the Appendix A.3 Model Architecture.
>
> **Significance**
>
> The ICLR call for papers explicitly lists applications in neuroscience as a relevant topic for the conference. Our paper describes significant contributions to the modeling of neural circuits using connectomic constraints. Our work combines biological realism with variational methods to test a variety of important mechanistic model classes against real data. Most excitingly, our work demonstrates the machine learning machinery required for using connectomic constraints to make specific detailed predictions about the neural activity of unmeasured neurons, which has not been possible with previous data-driven statistical modeling techniques.

---

### Official Review · Reviewer_UhCh · 2021-11-02

**Correctness:** 3
**Technical Novelty And Significance:** 2
**Empirical Novelty And Significance:** 3
**Recommendation:** 8
**Confidence:** 3

**Main Review:**

This paper makes some additional developments on the C. elegans latent variable modeling literature: including both chemical and electrical synapse information, adding conductance-based synapses, and nonlinear calcium observations with a time constant. However, I don’t believe it’s the first linear latent variable model to consider connectomic constraints (or fitting linear models to C elegans data with variational inference). See equation 1 in Mena et al (2017). Also see Linderman et al (2019) for cross-worm comparisons and predictions of held-out neurons.

The clarity of the methods and conclusions could be improved.
Methods;
-Does 'g' in equation 5 denote the softplus? It's not clear whether the 'same' in the sentence before refers to g being the same across neurons, same as the softplus definition earlier in the section or both.
	-Is there a reference supporting this observation model? Is the threshold of 'g' fixed?

-Page 4 section 2.1 redundantly refers to Bargmann et al about nonspiking neurons in C. elegans twice in the first two paragraphs. I'd suggest moving this reference earlier to the introduction where you first mention that you're using a nonspiking model as many in the ICLR audience may not know this fact and it'd more quickly support your modeling assumptions.

-Last paragraph of page 3 "A latent variable model....". I found this paragraph a little confusing. I wasn't sure what many sentences were referring to, especially with the introduction of the "black-box" networks (2nd sentence vs the black-box CCN brought up in the third sentence).

-The explanation of variational inference over SMC (top text on page 3, starting “inference instead of SMC”) was a little unclear what the authors wanted to convey with “generates a different type of distribution and is more computationally efficient because it allows direct sampling from the approximate posterior.” A more explicit definition of “different” would help clarity (or removing this altogether) and it’s also unclear whether sampling from the posterior is used to help validate/apply the model here.

Conclusions:
-The conductance-based synapses showed a performance improvement in the neuron holdout, but not really in the worm holdout condition. Why is this and does it impact the conclusions?

-How does the calcium observation model impact the model fit? This is an important contribution of the proposed model, but it's not clear if the Ca dynamics are necessary compared to, for instance, a simple linear-Gaussian model (given the observation window size).

**Summary Of The Paper:**

The authors propose a biologically constrained latent linear dynamical model of the C. elegans nervous system. They use connectomic information (including chemical vs electrical synapses) to constrain connections between units during inference. They fit the model to calcium imaging data from whole C. elegans using a variational approach. They find that biological constraints improve model performance and validity using both withheld neuron and across worm metrics.

**Summary Of The Review:**

The paper highlights that known biological constraints can be included tractably within latent variable models of biological neural networks (in this case, the C. elegans). Moreover, they provide some evidence that these constraints improve model fits.
The model goes beyond existing linear latent variable models of C. elegans. However, the conclusions do not include strong measures comparing against existing methodology and it's difficult to gauge how much advancement this model provides. While the presentation clarity was adequate, organization and model presentation could be improved.

---

> ### Author Response · Authors · 2021-11-23
> **Thank you for your thoughtful review of our paper. Answers related to linear dynamics model, related works, softplus, neuron v.s. worm holdout, calcium observation model.**
>
> Thank you for your thoughtful review of our paper. We have significantly revised our paper based on your suggestions, and we additionally clarify some details below.
>
> **Not a linear dynamical model**
>
> Firstly, please note that our model is not a “latent **linear** dynamical model”, as you say. We have developed a mechanistically realistic network model for the C. elegans nervous system. Since most neurons in the worm are believed to be non-spiking, we modeled the voltage dynamics of individual neurons with passive linear dynamics. However, we developed two nonlinear models for the chemical synapses between neurons (current, conductance). Further, our electrical synapse models are also nonlinear. Thus the full differential equations for the network dynamics are distinctly nonlinear in two different ways. Indeed the nonlinear dynamics are a major reason for needing to perform approximate variational inference.
>
> **Relationship to Mena 2017, Linderman 2019, blackbox vs mechanistic models**
>
> Mena 2017, and Linderman 2018, do use weak connectomic information to constrain a linear dynamical model of the C. elegans nervous system. However, they do not distinguish between electrical and chemical synapses, and do not model them in a realistic manner. Further, they did not characterize the effectiveness of the connectome in making experimentally/biologically relevant predictions about the activities of unmeasured neurons.
> In contrast, our work quantifies the usefulness of connectomic constraints to make experimentally testable predictions about the activities of unrecorded neurons, and further compares several families of mechanistic neural network models to discover the best way to make use of the connectomic information.
> Finally, the generative model in Linderman 2019 is not a mechanistic connectome based neural network model, but rather an abstract low-dimensional dynamical system. We refer to such statistical models as **blackbox** because the generative model does not have a correspondence to real neuron in the nervous system and their connectivity. Such a model can capture the correlations present in the measured data, but cannot make inferences about unmeasured neurons. Please note that the neuron hold out analysis in Linderman 2019 is not the same as our neuron hold out. In Linderman 2019, the model is always trained using neurons which are later withheld at test time. In our paper, neurons are withheld both at training time and test time. Withheld neurons are treated as missing data, and are missing at all times, both training and test. Linderman 2019 cannot model this scenario – this model can never make predictions about neurons which were never measured.
>
> **'g'=softplus?**
>
> ‘g’ represents a softplus with a fixed threshold of 20 in equation 5. The function of equation 5 is the same across neurons. All ‘g’ in our paper represents a softplus activation with a fixed threshold 20. We have included clarifications in our revision.
>
> **Neuron holdout vs worm holdout**
>
> The neuron holdout experiments are the strongest test of the quality of our model. These experiments show how well the model constraints unmeasured neurons, which are essentially hidden neurons. Correctly predicting the activity of unmeasured neurons suggests that the model is correctly inferring the mechanism by which the activity of the measured neurons is generated. In contrast, the worm holdout experiments that show good performance across all models. This shows that it is possible to fit the measured neural activity equally well with and without connectomic constraints – this demonstrates how it is challenging to assess the quality of a model at making causal predictions from the worm holdout experiments.
>
> **Calcium observation model**
>
> We hypothesized that the known connectome would only generate the correct voltage dynamics if simulated sufficiently accurately. Hence it would be necessary to have fast voltage dynamics, and slow calcium dynamics to match the observations. For this reason, we built a novel nonlinear voltage to calcium dynamics model for non-spiking neurons (traditional calcium models are based on calcium kernels acting on single action potentials). We build our model with a leaky integration model for the calcium observation model. We model the ion concentration with a softplus activation of neuron voltage because ion-concentration should be nonnegative and naturally decays over time as the membrane potential decreases. A linear gaussian model would not capture these dynamics.

---

> > ### Comment · Reviewer_UhCh · 2021-11-30
> > **Thanks for the clarifying comments**
> >
> > I thank the authors for their extensive replies.
> >
> > I apologize for the mislabeling of the model as linear - that was a slip of the tongue as I was mostly excited about the difference in chemical/electrical synapses in this paper which isn't linear.
> >
> > For the "Calcium observation model"
> > I understand why the authors included this in the model. What my question was trying to get at is what the effect of including this was on the model fit and conclusions. Including some quantification of this (model fitness with a simple linear Gaussian vs the more complex nonlinear activity) would improve the impact of this paper on how people model these types of observations. This isn't a huge sticking point for me, but I think it's important to consider all the biophysical factors included given the paper's motivation.
> >
> > The worm vs neuron holdout differences make more sense now. I think this section has the potential to be much more clear given your detailed conversation with another reviewer.

---

### Official Review · Reviewer_5GJS · 2021-11-02

**Correctness:** 3
**Technical Novelty And Significance:** 2
**Empirical Novelty And Significance:** 3
**Recommendation:** 8
**Confidence:** 4

**Main Review:**

The question of whether anatomical constraints help with improving model predictions and constraining the space of generative models of neural activity is an important and significant question in computational neuroscience. This paper takes a step towards addressing this question by developing biological models that have interpretable parameters (connectivity weights, neuron time constants, resting membrane potentials) yet are flexible to capture the variability across the population or individual neurons (by incorporating noise in the latent and observed variables). If a model resembles the biological generative process of the data up to appropriate amount of details, we would expect that incorporating more biological constraints would help with model's predictive performance. Therefore this approach not only could be used for model selection, but also could provide insight about why biological systems prefer certain architectures or mechanisms over others.

While the paper is clear and interesting, there are several issue that need more clarification as listed below.

* [Hierarchical Bayesian models](https://www.biorxiv.org/content/10.1101/621540v1.article-info) are already developed to study the dynamics in *C. elegans*. How would the results change if the anatomical constraints are imposed on the already established models such as the one mentioned above. Some comparison with this paper would make the claims stronger.
* In the results, why neuron-held-out correlation is worse than worm-held-out? Am I misunderstanding this? At least intuitively shouldn't we expect to have better performance when a single neuron is missing compared to all the neurons from a worm missing?
* How did the authors perform neuron-hold-out experiments? If a neuron is missing in the training dataset, then how do we have a latent variable corresponding to it? Is the missing neuron treated as missing observations? At the test time, do we use the "reconstruction" for evaluating the correlations? Is the reconstruction the mean of the posterior or is it based on samples?
* For the neuron-hold-out experiments, pairs of neurons (107) are considered. Why is there 107 pairs and why are we considering pairs of neurons instead of removing a single neuron? Again I feel like I'm misunderstanding the neuron-hold-out experiments.
* Is it possible to visualize the reconstructed activity of all the neurons (for a hold-out-worm and training worm) in the appendix? This would be important for biology experts and can provide further insights into the proposed model.
* How should we ensure that the model is learning time dynamics? The factorized Gaussian family does not incorporate any dynamical model or smoothness constraints in my understanding, is that right?
* When using anatomical constraints for model selection we need to control and account for as many confounding factors as possible that can potentially lead to misinterpretations. These factors include (but not limited to) controlling for the number of parameters across different models (more parameters can lead to overfitting, can make the optimization harder, can make the inference harder since it's in a higher dimensional space), and successful training (what is the convergence criterion?).
* In Fig. 3 the error bars seem too large, are the differences significant in the violin plots? In addition, some neurons have really bad predictive performance, can we focus on those neurons to understand why this is happening? Is it due to lack of measurements form their neighboring neurons? Is it because they are less predictable and more impacted by unmeasured sources of variability?


I have some minor comments to, and would like the authors to clarify.

* What is $\theta$? Please write down $\theta$ explicitly using the notations introduced in the paper.
* According to Wikipedia, *C. elegans* has 302 neurons, not 300. Can you clarify this?
* What is $\phi$? Are you using a neural network to parametrize $\phi$? If so what kind of architecture is used?
* How does the regularization work? What is the probabilistic interpretation of adding the $|| |M| -|C| ||$ regularization? Is this equivalent to defining a normal prior? Please clarify this. Why did the authors use $L_2$ regularization? Isn't $L_1$ regularization associated with sparsity? If so, what is the probabilistic interpretation of this in the variational inference framework?






**Summary Of The Paper:**

The paper develops a latent variable model with biologically meaningful parameters for the worm *C. elegans* whole brain neural traces. Given the connectivity between the neurons, the neural traces are modeled as stochastic leaky integrators with either conductance or current based synaptic input model. Given the voltage traces, the observed calcium signals are modeled as first order leaky integrators added by noise to account for the observation noise. External input is nonlinearly transformed and fed into the neurons to account for the stimulus-dependent modulation of neural activity. Multiple connectivity models (fully trainable, fully trainable with $L_2$ regularization, trainable weight magnitudes, trainable global scale) are used to investigate whether biological connectome constraints help with neural activity predictions or not. Variational inference is used to infer a posterior distribution over neural activity trajectories with factorized Gaussian distribution as the variational family. Results on neuron-hold-out and worm-hold-out experiments suggest that connectome constraints and conductance-based modeling improves model prediction over unseen neurons and worms.

**Summary Of The Review:**

While the approach is very interesting and the problem considered is important, the paper lacks comparisons and clearer description and investigation of the results. Furthermore, some justification on specific choices need to be done. See above for more detailed comments.

Post discussion: I'm increasing my score to "accept" since most of my comments are incorporated, I thank the authors for their hard work in revising the paper and incorporating additional experiments/visualizations.

---

> ### Author Response · Authors · 2021-11-23
> **Thank you for your careful review. Answers related to relationship to hierarchical bayesian models, neuron-hold out vs worm-hold out, neuron pairs clarification, more visualizations, temporal dynamics, overparameterization.**
>
> Thank you for your careful review. Thanks to your feedback, we have significantly revised our paper. We also address your comments below.
>
> **Relationship to Hierarchical Bayesian models, significance of neuron-hold out prediction**
>
> Linderman 2019 builds a low-dimensional probabilistic state space model of the calcium dynamics of head ganglion neurons in C. elegans. Such a black-box statistical model can fit the correlations in the measured data, but it is not a mechanistic model and cannot predict the activity of neurons which were never measured. In contrast, we have built a mechanistic model of the voltage dynamics of the complete worm nervous system based on the complete connectome and partial activity measurements.
> Our mechanistic model has the ability to predict the activities of neurons which were never experimentally measured. Because neural voltage activity in our model can only be generated by the voltage dynamics of other neurons in the nervous system through synaptic couplings dictated by the connectome, even calcium imaging experiments of just a subset of the nervous system can help constrain the activities of unrecorded neurons. Only by virtue of the connectome is it possible to make such predictions.
> We experimentally demonstrated our ability to make predictions about unmeasured neurons using the neuron hold out paradigm. We fit a version of the model while holding out a pair of bilaterally symmetric neurons – by pretending that these neurons were never measured. And we ask whether that model can correctly predict the activity of these withheld neurons.
> Please note that this is different from the more classic neuron hold out paradigm, where the model is fit using data that includes the withheld neurons, and the neurons are withheld only at test time.
> Withheld neurons and all other unobserved neurons are treated as missing observations but all neurons are always modeled with latent variables. The voltage trace of the held-out neuron just does not get fit against data in the reconstruction loss. The reconstruction is used to evaluate the correlations and is from the mean of the posterior.
>
> **Neuron-hold out vs worm-hold out**
>
> In Neuron-hold-out, the models are trained on one worm but the observation of an individual neuron (or bilateral neuron pair) is removed from the training data, while missed neurons are still modeled as latent variables. In Worm-hold-out, the models are trained on some of the worms but not others. However, the training data contains observations from all the observed neurons. The worm hold-out predictions can be better because they see the predictions of the individual neurons from the data from other worms and are able to generalize to a separate animal, while in neuron-hold-out, this neuron is never seen by the model, neither in training nor test. During test, we used the reconstructed fluorescence trace and measured ground truth for evaluation. We have added more clarifications in Sec. 3.2 , 3.3 in our revison.
>
> **"Why are there 107 pairs..."**
>
> Most neurons in C. elegans are bilaterally symmetric (White 1986) and the symmetric neurons tend to have highly correlated activities. We evaluate the neurons in symmetric pairs in order to prevent the model from predicting the activity of one neuron purely based on its symmetric twin. However, not all the neurons have a symmetric pair: these singlet neurons were held out individually.
>
> **More visualizations**
>
> We have included visualizations for all neurons in one train worm and one hold-out worm in Appendix A.11 in our revision.
>
> **Temporal dynamics**
>
> The prior distribution over voltage trajectories is described by the mechanistic connectome constrained nonlinear voltage dynamics described by our generative model. This prior distribution is what ensures that the model learns the correct temporal dynamics, via the KL divergence term in the ELBO objective function. Our approximate variational posterior distribution is indeed a factorized gaussian approximation. This does not prevent the posterior from correctly modeling temporal dynamics. But the approximation does imply that we do not necessarily capture any correlations in the posterior across time or neurons.
>
> **Overparameterization, over/under-fitting**
>
> It is true that having an overparameterized model can hurt performance. This raises the question of whether the improved performance of the connectome-constrained model over that of the learned-dense (fully connected, unconstrained) model is actually due to the connectome constraint. For this reason, we have tried fitting a learned-sparse (sparsely connected, unconstrained) model, in which the connections between neurons are learned but are forced to be sparse so that the number of parameters in the system is limited. We found that the learned sparse model still performs substantially worse than our CC-LVM which implies that the improvement in performance is not solely attributed to less overparameterization.

---

> > ### Author Response · Authors · 2021-11-23
> > **Thank you for your careful review. Answers related to error bars, theta/phi, 300 v.s. 302 neurons, regularization.**
> >
> > Thanks to your feedback, we have significantly revised our paper. We also address additional comments below.
> >
> > **Fig. 3 “error bars”**
> >
> > Violin plots represent the distributions of neuron-hold-out predictive performance of Individual neurons in each neuron category. They don’t show error bars.  Some neurons that have lower correlations are due to less constraints from upstream and downstream neurons, as well as more complex trajectories.
> >
> > **“What is theta? phi?.”**
> >
> > We have added this information to the end of Appendix A.5 in the appendix.
> >
> > **300 neurons? 302 neurons?**
> >
> > There are 2 neurons (CAN cells) not connected to the rest of the whole-brain; the public connectome in most studies provide a connectivity matrix with a size of 300x300 (Witvliet 2021). Early developmental work identified CAN cells as neurons because they appear anatomically similar, but we now believe they do not function as neurons in the adult.
> >
> > **Regularization**
> >
> > We have clarified the regularizations used in our paper. We now introduce two different mechanisms for regularizing the “learned connectomes”, which are our baseline comparisons. First is a standard L1 sparsifying regularization (“learned sparse”). Second is a regularizer that attempts to match the total number of synapses between the true connectome and the “learned connectome” (“learned total count”). The updated results are reported in Table 2 and Figure 3 in our revision. These results are still consistent to our conclusion.

---

### Official Review · Reviewer_iieb · 2021-11-03

**Correctness:** 3
**Technical Novelty And Significance:** 3
**Empirical Novelty And Significance:** 3
**Recommendation:** 6
**Confidence:** 4

**Main Review:**

•	I believe that connectome constraints help improve the model performance in the current setup. However, the results are not surprising. The question is whether the benefit of connectome constraints was due to insufficient training data. When you have enough data, would an unconstrained model or a loosely constrained model perform as well as a connectome constrained model, meaning that a model could learn reasonable connection structure through sufficient training?

•	How does the latent space look like? The current model predicted the latent variable of neuron voltage and used a differential equation to further predict the observed calcium signal. One major question I had was whether such a simplified calcium-voltage model was sufficient for C.elegans neurons. The units of model networks were passive non-spiking units, while in real brain calcium signal was mainly generated by suprathreshold neuronal activities. Although the problem might be partially taken care of by the nonlinearities of the model, it is not clear whether that was sufficient. On the other hand, I’m wondering whether the oversimplified calcium-voltage model could be one reason causing the relatively worse single neuron prediction performance of sensory neurons. Considering the spiking properties of sensory neurons, they might be more sensitive to inaccurate calcium-voltage models compared to interneurons and motor neurons.

•	How does warm prediction look like? Could authors show some population trajectories? Could authors discuss what had led to the inaccuracy in whole warm predictions, whether due to inaccurate prediction of a particular subset of neurons or whether due to inaccurate prediction of overall brain state?


**Summary Of The Paper:**

The current manuscript presented a connectome constrained latent variable model for whole-brain calcium imaging of C.elegans nervous system. The whole-brain calcium imaging of C.elegans was collected while C.elegans was undergoing chemosensory testing, and the dataset has been published recently. In the current study, the authors aimed to present a model that could predict the single-neuron and the single-trial activity of this dataset. Specifically, the activity of each neuron of the whole brain imaging was modeled by latent variable analogous to the voltage of one unit in the model network. The connection between model network units was constrained by the connectome. The authors showed that the connectome constraints significantly improved the prediction power of the model, in predicting the activity of missing units, as well as predicting the single-trial activity of hold-out warms. Overall, the authors have provided a clear description of the model and carried out a good amount of experiments to evaluate different model variants.

**Summary Of The Review:**

In general, I vote for accepting the current paper if the authors could address all of my concerns above. I think the authors have presented a model for a novel whole-brain calcium imaging which was only recently published. Overall, the modeling strategy makes sense. I like that the authors tested on different variants of the model, which are all relevant, and provided insights on model selection. One limitation of the current work is the model generalization. I think the application of the model is limited to the current training data. I’m not convinced that the current model could generalize to chemosensory tasks, other than the ones in the training sets.  My major concern is about the clarity of some parts of the paper and some additional improvement in the work. I hope the authors could address my concerns in the rebuttal.

---

> ### Author Response · Authors · 2021-11-23
> **Thank you for your thoughtful review. Answers related to benefit of connectome constraints, latent space visualization, simplified model, prediction or sensory neurons & model generalization.**
>
> Thank you for your thoughtful review. We have revised our paper based on your feedback, and address your review below.
>
> **Benefit of connectome constraints**
>
> The NeuroPAL-based calcium imaging dataset (Yemini 2021) is one of the first datasets to measure the activity of a significant fraction of the C. elegans nervous system simultaneously, with high confidence cell identity. Yet even this dataset does not contain activity measurements for all neurons. Our work shows that a complete connectome can be combined with incomplete calcium imaging to build a mechanistic model for the entire nervous system. This is demonstrated by our neuron-hold out experiments.
> Further, even if complete activity measurements were available, in the absence of connectomic constraints, there can be multiple networks capable of generating the same neural activity. Thus from activity (correlations) alone, it is not possible to infer mechanism (causality).
>
> **Latent space visualization, population trajectories**
>
>  We have added predicted voltage traces from worm holdout and latent space visualization to Appendix A.9, A.10, A.11 in our revision. We find the inaccuracy in whole warm predictions are due to inaccurate prediction of overall brain state.
>
> **Passive voltage dynamics, simplified calcium-voltage model sufficient?**
>
> While recent work has uncovered action potentials in some neurons in C. elegans (Mellem 2008, Liu 2018), it is still believed that the majority of neurons in the C. elegans nervous system are non-spiking (Bargmann 1998, Goodman 1998, Lockery 2009). Thus, we believe our passive dynamics model for the neuronal voltage to be a reasonably good approximation of reality. Classic synapse models, and classic calcium models typically assume spiking neurons, and generate PSPs or calcium events in response to individual action potentials. Since our neurons are believed to be non-spiking, we had to develop new nonlinear voltage to calcium and voltage to synaptic current models. Our results suggest that our novel synapse and calcium models are a reasonable approximation to reality. The exploration of richer neuron, synapse, and calcium models will be an important future direction for research.
>
> **Poor prediction of sensory neurons**
>
> While the connectome constrains the neuron to neuron coupling, it does not reveal the coupling of sensory neurons to the environment. We instead had to learn a sensory mapping from odors to sensory neurons, but this is very likely an incomplete description of the worm’s sensory environment. We hypothesize that the poor prediction of sensory neurons is due to missing measurements of the sensory inputs.
>
> **Model generalization**
>
> We agree that it is too early to know the degree to which a single model can generalize to a variety of experimental conditions and datasets. As new datasets become available, this will be a key area for future model development.

---

### Decision · Program_Chairs · 2022-01-20

**Decision:**

Accept (Poster)

**Comment:**

The authors build an encoding model of whole-brain brain activity by integrating incomplete functional data with anatomical/connectomics data. This work is significant from a computational neuroscience perspective because it constitutes a proof of concept regarding how  whole brain calcium imaging data can be used to constrain the missing parameters of a connectome-constrained, biophysically detailed model of the C. elegans nervous system. There were issues related to clarity in the initial submission which all appeared to have been addressed in the final revision. This paper received 3 accepts (including one marginal accept) and 1 reject. The paper was discussed and the reviewers (including the negative reviewer) were unanimous that the current submission should be accepted.